# Provably Adversarially Robust Detection of Out-of-Distribution Data (Almost) for Free

**Alexander Meinke**[*]
University of Tübingen
Tübingen AI Center

**Julian Bitterwolf**
University of Tübingen
Tübingen AI Center

**Matthias Hein**
University of Tübingen
Tübingen AI Center

## Abstract

The application of machine learning in safety-critical systems requires a reliable assessment of uncertainty. However, deep neural networks are known to produce highly overconfident predictions on out-of-distribution (OOD) data. Even if trained to be non-confident on OOD data, one can still adversarially manipulate OOD data so that the classifier again assigns high confidence to the manipulated samples. We show that two previously published defenses can be broken by better adapted attacks, highlighting the importance of robustness guarantees around OOD data. Since the existing method for this task is hard to train and significantly limits accuracy, we construct a classifier that can simultaneously achieve provably adversarially robust OOD detection and high clean accuracy. Moreover, by slightly modifying the classifier's architecture our method provably avoids the asymptotic overconfidence problem of standard neural networks. We provide code for all our experiments.[†]

## 1 Introduction

Deep neural networks have achieved state-of-the-art performance in many application domains. However, the widespread usage of deep neural networks in safety-critical applications, e.g. in healthcare, autonomous driving/aviation, manufacturing, raises concerns as deep neural networks have problematic deficiencies. Among these deficiencies, overconfident predictions on non-task related inputs [44, 21] have recently attracted a lot of interest. Even theoretically derived weaknesses like ReLU networks provably being overconfident far away from the training data [20] are yet to be fixed. Meanwhile, reliable confidences of the classifier on the classification task (in-distribution) [19] as well as on the out-distribution [21, 20] are important to be able to detect when the deep neural network is working outside of its specification, which can then be used to either involve a human operator or to fall back into a "safe state". Thus, solving this problem is of high importance for trustworthy ML systems. Crucially, a detection method needs to generalize to novel test out-distributions that are not available during training, since one does not know which unknown inputs can be expected.

Many approaches have been proposed for OOD detection, [21, 33, 31, 32, 22, 48, 20, 41, 9, 46, 35, 36]. In this paper we focus on confidence based OOD detection, i.e. the probability of the predicted class is used to decide whether to reject or accept the sample, because of its straightforward interpretation and because it has been shown to perform no worse than other scores for OOD detection [7]. One of the currently best performing methods enforces low confidence during training ("outlier exposure" (OE)) on a large and diverse set of out-distribution images [22] which leads to strong separation of in- and out-distribution based on the confidence of the classifier.

---

[*]Corresponding author - `alexander.meinke@uni-tuebingen.de`

[†]`https://github.com/AlexMeinke/Provable-OOD-Detection`

Table 1: **ProoD combines desirable properties of existing (adversarially robust) OOD detection methods.** It has high test accuracy and standard OOD detection performance (as [22]) and has worst-case guarantees if the out-distribution samples are adversarially perturbed in an $l_\infty$-neighborhood to maximize the confidence (see Section 4.2). Similar to CCU [41] it avoids the problem of asymptotic overconfidence far away from the training data.

| | OE [22] | CCU [41] | ACET/ATOM [20, 9] | GOOD [8] | ProoD |
|---|---|---|---|---|---|
| High accuracy | ✓ | ✓ | ✓ | | ✓ |
| High clean OOD detection performance | ✓ | ✓ | ✓ | | ✓ |
| Adv. OOD $l_\infty$-robustness | | | (✓) | ✓ | ✓ |
| Adv. OOD $l_\infty$-certificates | | | | ✓ | ✓ |
| Provably not asympt. overconfident | | ✓ | | | ✓ |

A remaining robustness problem of standard OOD detection methods is that they are vulnerable to adversarial perturbations, i.e. small modifications of OOD inputs can lead to large confidence of the classifier on the manipulated samples [44, 20, 50]. Of course, an OOD input, which by definition is semantically far away from the in-distribution, should not be able to be moved into a region that is considered in-distribution by the detection model if the movement is imperceptibly small. On the other hand, a slightly perturbed in-distribution input can still be considered in-distribution for some perturbations (e.g. if the perturbation resembles standard camera noise), but for other perturbations it might be highly atypical and therefore should arguably rather be seen as OOD. Furthermore, adversarial robustness on the in-distribution is known to come at the cost of clean accuracy [53] which hinders the adoption of such methods in practice. We aim to provide a method that does not harm the in-distribution performance in any way and thus, like previous OOD-detection methods, we do *not* consider adversarially manipulated in-distribution samples and focus on ensuring that OOD samples remain OOD under adversarial attacks.

While different methods for adversarially robust OOD detection have been proposed [20, 50, 41, 9, 8] there is little work on *provably* adversarially robust OOD detection [41, 8, 25, 5]. For the standard empirical evaluation of adversarial robustness, for each input one runs an array of different attacks that conform to the assumed threat model and records the output for the worst found perturbation. This means that there is no guarantee that a more malign perturbation does not exist, as only a lower bound on the adversarial robustness is established. Provable adversarial robustness on OOD data, which we provide in this paper, yields a mathematically deduced upper bound on the worst-case confidence around each OOD sample. For our guaranteed upper bounds on the confidence of an OOD sample, it is certain that no applicable manipulation raises the confidence above the certified value.

In [25] they apply randomized smoothing to obtain guarantees wrt. $l_2$-perturbations for Dirchlet-based models [39, 40, 51] which already show quite some gap in terms of AUC-ROC to SOTA OOD detection methods even without attacks. Interval bound propagation (IBP) [17, 42, 56, 23] has been shown to be one of the most effective techniques in certified adversarial robustness on the in-distribution when applied during training. In GOOD [8] they use IBP to compute upper bounds on the confidence in an $l_\infty$-neighborhood of the input and minimize these upper bounds on a training out-distribution. This yields classifiers with pointwise guarantees for adversarially robust OOD detection even for "close" out-distribution inputs which generalize to novel OOD test distributions. However, the employed architectures of the neural network are restricted to rather shallow networks as otherwise the bounds of IBP are loose. Thus, they obtain low classification accuracy which is far from the state-of-the-art, e.g. 91% on CIFAR10, and their approach does not scale to more complex tasks like ImageNet. In particular, despite its low accuracy the employed network architecture is quite large and has higher memory consumption than a ResNet50. The authors of [5] use SOTA verification techniques [15] and get guarantees for OOD detection wrt. $l_\infty$-perturbations for ACET models [20] that were not specifically trained to be verifiable but the guarantees obtained by training the models via IBP in [8] are significantly better.

A *different* type of guaranteed low confidence on OOD data pertains to the asymptotic behavior of a classifier. Since standard ReLU networks provably have increasing confidence in almost all directions far away from the training data [20], one has to modify the architecture in order to solve this problem. In CCU [41] the authors append density estimators based on Gaussian mixture models for in- and out-distribution to the softmax layer. By also enforcing low confidence on a training out-distribution, they achieve similar OOD detection performance to [22] but can guarantee that the classifier shows

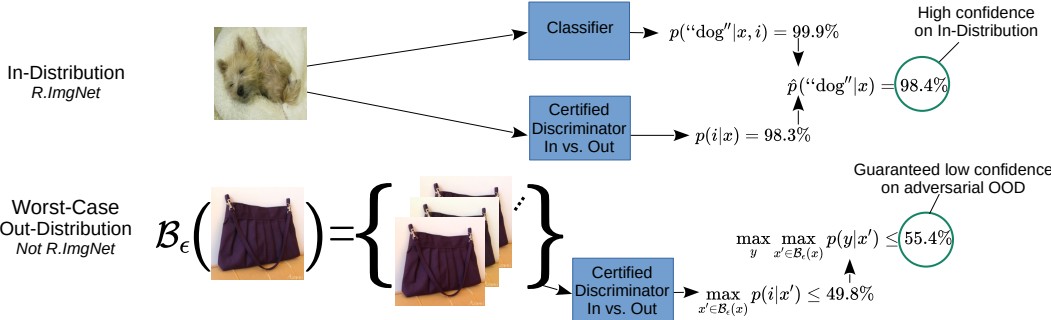

Figure 1: **ProoD's Architecture:** Combining the output of a classifier and a certified discriminator, see Eq. (2), we achieve high confidence on the in-distribution sample of a dog (R.ImgNet). The certified discriminator, see Eq. (4), yields an upper bound on the confidence in a $\ell_\infty$-neighborhood of the shown OOD sample not belonging to any classes of R.ImgNet. ProoD achieves provable guarantees on adversarial OOD detection without loss in accuracy or clean OOD detection.

decreasing confidence as one moves away from the training data. However, for close in-distribution inputs this approach yields no guarantee as the Gaussian mixture models are not powerful enough for complex image classification tasks. In [27, 28] similar asymptotic guarantees are derived for Bayesian neural networks but without any robustness guarantees.

In this paper we propose ProoD which merges a certified binary discriminator for in-versus out-distribution with a classifier for the in-distribution task in a principled fashion into a joint classifier. This combines the advantages of CCU [41] and GOOD [8] without suffering from their downsides. In particular, ProoD simultaneously achieves the following:

- Guaranteed adversarially robust OOD detection via certified upper bounds on the confidence in $l_\infty$-balls around OOD samples.

- Additionally, it provably prevents the asymptotic overconfidence of deep neural networks.

- It can be used with arbitrary architectures and has no loss in prediction performance and standard OOD detection performance.

Thus, we get provable guarantees for adversarially robust OOD detection, fix the asymptotic over-confidence (almost) for free as we have (almost) no loss in prediction and standard OOD detection performance. We qualitatively compare the properties of our model to prior approaches in Table 1.

## 2 Provably Robust Detection of Out-of-distribution Data

In the following we consider feedforward networks for classification, $f : \mathbb{R}^d \to \mathbb{R}^K$, with $K$ classes defined with input $x^{(0)} = x$ and layers $l = 1, \ldots L - 1$ as

$$x^{(l)} = \sigma^{(l)}\left(W^{(l)}x^{(l-1)} + b^{(l)}\right), \qquad\qquad f(x) = W^{(L)}x^{(L-1)} + b^{(L)}, \qquad (1)$$

where $W^{(l)}$ and $b^{(l)}$ are weights and biases and $\sigma^{(l)}$ is either the ReLU or leaky ReLU activation function of layer $l$. We refer to the output of $f$ as the *logits* and get a probability distribution over the classes via $\hat{p}(y|x) = \frac{e^{f_y(x)}}{\sum_k^K e^{f_k(x)}}$ for $y = 1, \ldots, K$. We define the confidence as $\text{Conf}(f(x)) = \max_{y=1,\ldots,K} \hat{p}(y|x)$.

### 2.1 Joint Model for OOD Detection and Classification

In our joint model we assume that there exists an in- and out-distribution where the out-distribution samples are unrelated to the in-distribution task. Thus, we can formally write the conditional distribution on the input as

$$\hat{p}(y|x) = \hat{p}(y|x, i)\hat{p}(i|x) + \hat{p}(y|x, o)\hat{p}(o|x), \qquad (2)$$

where $\hat{p}(i|x)$ is the conditional distribution that sample $x$ belongs to the in-distribution and $\hat{p}(y|x, i)$ is the conditional distribution for the in-distribution. We assume that OOD samples are unrelated and thus maximally un-informative to the in-distribution task, i.e. we fix $\hat{p}(y|x, o) = \frac{1}{K}$, so that the classifier can be written as

$$\hat{p}(y|x) = \hat{p}(y|x, i)\hat{p}(i|x) + \frac{1}{K}(1 - \hat{p}(i|x)). \tag{3}$$

We train the binary classifier $\hat{p}(i|x)$ in a certified robust fashion wrt. an $l_\infty$-threat model so that even adversarially manipulated OOD samples are detected. In order to avoid confusion with the multi-class classifier, we will refer to $\hat{p}(i|x)$ as a binary discriminator. In an $l_\infty$-ball of radius $\epsilon$ around $x \in \mathbb{R}^d$ and for all $y$ we get the upper bound on the confidence of the final classifier in Eq. (3):

$$\max_{\|x'-x\|_\infty \leq \epsilon} \hat{p}(y|x') \leq \max_{\|x'-x\|_\infty \leq \epsilon} \hat{p}(i|x') + \frac{1}{K}\left(1 - \hat{p}(i|x')\right) = \frac{K-1}{K} \max_{\|x'-x\|_\infty \leq \epsilon} \hat{p}(i|x') + \frac{1}{K}, \tag{4}$$

where we have used that $p(y|x, i) \leq 1 \forall x, y$, so we can defer the certification "work" to the binary discriminator. Using a particular constraint on the weights of the binary discriminator, we get similar asymptotic properties as in [41] but additionally get certified adversarial robustness for close out-distribution samples as in [8]. In contrast to [8], this comes without loss in test accuracy or non-adversarial OOD detection performance since in our model the neural network used for the in-distribution classification task $\hat{p}(y|x, i)$ is independent of the binary discriminator. Thus, we have the advantage that the classifier can use arbitrary deep neural networks and is not constrained to certifiable networks. We call our approach **Pr**ovable **o**ut-**o**f-**D**istribution detector (ProoD) and visualize its components in Figure 1. The intuitive idea of why ProoD can achieve adversarially robust OOD detection without loss in clean OOD detection can be explained with the behavior of the predicted probability distribution provided in Equation (3).

- **For clean OOD:** the classifier $\hat{p}(y|x, i)$ (trained similar to Outlier Exposure) already enforces low confidence on out-of-distribution points and thus irrespective of the values $\hat{p}(i|x)$, the resulting output of $\hat{p}(y|x)$ will be close to uniform as well and thus ProoD performs similar to Outlier Exposure.

- **For adversarial OOD:** the classifier confidence $\max_y \hat{p}(y|x, i)$ is potentially corrupted but now the binary discriminator $\hat{p}(i|x)$ kicks in and ensures that the resulting prediction $\hat{p}(y|x)$ is close to uniform.

This explains why the combination of certified discriminator and classifier works much better than the individual parts and the use of this "redundancy" is the key idea of ProoD.

**Certifiably Robust Binary Discrimination of In- versus Out-Distribution**    The first goal is to get a certifiably adversarially robust OOD detector $\hat{p}(i|x)$. We train this binary discriminator independently of the overall classifier as the training schedules for certified robustness are incompatible with the standard training schedules of normal classifiers. For this binary classification problem we use a logistic model $\hat{p}(i|x) = \frac{1}{1+e^{-g(x)}}$, where $g : \mathbb{R}^d \to \mathbb{R}$ are logits of a neural network (we denote the weights and biases of $g$ by $W_g$ and $b_g$ in order to differentiate it from the classifier $f$ introduced in the next paragraph). Let $(x_r, y_r)_{r=1}^N$ be our in-distribution training data (we use the class encoding $+1$ for the in-distribution and $-1$ for the out-distribution) and $(z_s)_{s=1}^M$ be our training out-distribution data. Then the optimization problem associated to the binary classification problem becomes:

$$\min_{\substack{g \\ W_g^{(L_g)} < 0}} \frac{1}{N} \sum_{r=1}^N \log\left(1 + e^{-g(x_r)}\right) + \frac{1}{M} \sum_{s=1}^M \log\left(1 + e^{\bar{g}(z_s)}\right), \tag{5}$$

where we minimize over the parameters of the neural network $g$ under the constraint that the weights of the output layer $W_g^{(L_g)}$ are componentwise negative and $\bar{g}(z) \geq \max_{u \in B_p(z,\epsilon)} g(u)$ is an upper bound on the output of $g$ around OOD samples for a given $l_p$-threat model $B_p(z, \epsilon) = \{u \in [0, 1]^d \mid \|u - z\|_p \leq \epsilon\}$. In this paper we always use an $l_\infty$-threat model. This upper bound could, in principle, be computed using any certification technique but we will use interval bound propagation (IBP) since it is simple, fast and has been shown to produce SOTA results [17]. Note that this is not standard adversarial training for a binary classification problem as here we have

an asymmetric situation: we want to be (certifiably) robust to adversarial manipulation on the out-distribution data but *not* on the in-distribution and thus the upper bound is only used for out-distribution samples. The negativity of the output layer's weights $W_g^{(L_g)}$ is enforced by using the parameterization $(W_g^{(L_g)})_j = -e^{h_j}$ componentwise and optimizing over $h_j$. In Section 3 we show how the negativity of $W_g^{(L_g)}$ allows us to control the asymptotic behavior of the joint classifier.

For the reader's convenience we quickly present the upper $\overline{x}^{(l)}$ and lower $\underline{x}^{(l)}$ bounds on the output of layer $l$ in a feedforward neural network produced by IBP:

$$\overline{x}^{(l)} = \sigma\left(W_+^{(l)}\overline{x}^{(l-1)} + W_-^{(l)}\underline{x}^{(l-1)} + b^{(l)}\right), \quad \underline{x}^{(l)} = \sigma\left(W_+^{(l)}\underline{x}^{(l-1)} + W_-^{(l)}\overline{x}^{(l-1)} + b^{(l)}\right), \quad (6)$$

where $W_+ = \max(0, W)$ and $W_- = \min(0, W)$ (min/max used componentwise). For an $l_\infty$-threat model one starts with the upper and lower bounds for the input layer $\overline{x}^{(0)} = x + \epsilon$ and $\underline{x}^{(0)} = x - \epsilon$ and then iteratively computes the layerwise upper and lower bounds $\overline{x}^{(l)}, \underline{x}^{(l)}$ which fulfill

$$\underline{x}^{(l)} \leq \min_{\|x'-x\|_\infty \leq \epsilon} x^{(l)}(x') \leq \max_{\|x'-x\|_\infty \leq \epsilon} x^{(l)}(x') \leq \overline{x}^{(l)}. \quad (7)$$

While in [8] they also used IBP to upper bound the confidence of the classifier this resulted in a bound that took into account all $\mathcal{O}(K^2)$ logit differences between all classes. In contrast, our loss in Eq. (5) is significantly simpler as we just have a binary classification problem and therefore only need a single bound. Thus, our approach easily scales to tasks with a large number of classes and training the binary discriminator with IBP turns out to be significantly more stable than the approach in [8].

**(Semi)-Joint Training of the Final Classifier**  Given the certifiably robust model $\hat{p}(i|x)$ for the binary classification task between in- and out-distribution, we need to determine the final predictive distribution $\hat{p}(y|x)$ in Eq. (2). On top of the provable OOD performance that we get from Eq. (4), we also want to achieve SOTA performance on unperturbed OOD data. In principle we could independently train a model for the predictive in-distribution task $\hat{p}(y|x, i)$, e.g. using outlier exposure (OE) [22] or any other state-of-the-art OOD detection method and simply combine it with our $\hat{p}(i|x)$. While this does lead to models with high OOD performance that also have guarantees, it completely ignores the interaction between $\hat{p}(i|x)$ and $\hat{p}(y|x, i)$ during training. Instead we propose to train $\hat{p}(y|x, i)$ by optimizing our final predictive distribution $\hat{p}(y|x)$. Note that in order to retain the guarantees of $\hat{p}(i|x)$ we only train the parameters of the neural network $f : \mathbb{R}^d \to \mathbb{R}^K$ and need to keep $\hat{p}(i|x)$ resp. $g$ fixed. Because $g$ stays fixed we call this semi-joint training. We use OE [22] for training $\hat{p}(y|x)$ with the cross-entropy loss and use the softmax-function in order to obtain the predictive distribution $\hat{p}_f(y|x, i) = \frac{e^{f_y(x)}}{\sum_k e^{f_k(x)}}$ from $f$:

$$\min_f -\frac{1}{N}\sum_{r=1}^N \log\left(\hat{p}(y_r|x_r)\right) - \frac{1}{M}\sum_{s=1}^M \frac{1}{K}\sum_{l=1}^K \log\left(\hat{p}(l|z_s)\right)$$

$$= \min_f -\frac{1}{N}\sum_{r=1}^N \log\left(\hat{p}_f(y_r|x_r, i)\hat{p}(i|x_r) + \frac{1}{K}\left(1 - \hat{p}(i|x_r)\right)\right)$$

$$- \frac{1}{M}\sum_{s=1}^M \frac{1}{K}\sum_{l=1}^K \log\left(\hat{p}_f(l|z_s, i)\hat{p}(i|z_s) + \frac{1}{K}\left(1 - \hat{p}(i|z_s)\right)\right), \quad (8)$$

where the first term is the standard cross-entropy loss on the in-distribution but now for our joint model for $\hat{p}(y|x)$ and the second term enforces uniform confidence on out-distribution samples. In App. B we show that semi-joint training leads to stronger guarantees than separate training.

The loss in Eq. (5) implicitly weighs the in-distribution and worst-case out-distribution equally, which amounts to the assumption $p(i) = \frac{1}{2} = p(o)$. This highly conservative choice simplifies training the binary discriminator but may not reflect the expected frequency of OOD samples at test time and in effect means that $\hat{p}(i|x)$ tends to be quite low. This typically yields good guaranteed AUCs but can have a negative impact on the standard out-distribution performance. In order to better explore the trade-off of guaranteed and standard OOD detection, we repeat the above semi-joint training with different shifts of the offset parameter in the output layer

$$b' = b_g^{(L_g)} + \Delta, \quad (9)$$

where $\Delta \geq 0$ leads to increasing $\hat{p}(i|x)$. This shift has a direct interpretation in terms of the probabilities $p(i)$ and $p(o)$. Under the assumption that our binary discriminator $g$ is perfect, that is

$$p(i|x) = \frac{p(x|i)p(i)}{p(x|i)p(i) + p(x|o)p(o)} = \frac{1}{1 + e^{-g(x)}}, \tag{10}$$

then it holds that $e^{g(x)} = \frac{p(x|i)p(i)}{p(x|o)p(o)}$. A change of the prior probabilities $\tilde{p}(i)$ and $\tilde{p}(o)$ without changing $p(x|i)$ and $p(x|o)$ then corresponds to a novel classifier

$$e^{\tilde{g}(x)} = \frac{p(x|i)\tilde{p}(i)}{p(x|o)\tilde{p}(o)} = \frac{p(x|i)p(i)}{p(x|o)p(o)} \frac{p(o)\tilde{p}(i)}{p(i)\tilde{p}(o)} = e^{g(x)}e^{\Delta} \tag{11}$$

with $\Delta = \log\left(\frac{p(o)\tilde{p}(i)}{p(i)\tilde{p}(o)}\right)$. Note that $\tilde{p}(i) > p(i)$ corresponds to positive shifts. In practice, this parameter can be chosen based on the priors for a particular application. Since no such priors are available in our case we determine a suitable shift by evaluating on the training out-distribution (see Section 4.2). Note that we explicitly do not train the shift parameter since this way the guarantees would get lost as the classifier implicitly learns a large $\Delta$ in order to maximize the confidence on the in-distribution, thus converging to a normal outlier exposure-type classifier without any guarantees.

## 3   Guarantees on Asymptotic Confidence

In this section we show that our specific construction provably avoids the issue of asymptotic overconfidence that was pointed out in [20]. Note that the resulting guarantee (as stated in Theorem 1) is different from and in addition to the robustness guarantees discussed in the previous section (see Eq. (4)). The previous section dealt with providing confidence upper bounds on neighborhoods around OOD samples whereas this section deals with ensuring that a classifier's confidence decreases asymptotically as one moves away from all training data.

We note that a ReLU neural network $f : \mathbb{R}^d \to \mathbb{R}^K$ as defined in Eq. (1) using ReLU or leaky ReLU as activation functions, potential max-or average pooling and skip connection yields a piece-wise affine function [2], i.e. there exists a finite set of polytopes $Q_r \subset \mathbb{R}^d$ with $r = 1, \ldots, R$ such that $\cup_{r=1}^R Q_r = \mathbb{R}^d$ and $f$ restricted to each of the polytopes is an affine function. Since there are only finitely many polytopes some of them have to extend to infinity and on these ones the neural network is essentially an affine classifier. This fact has been used in [20] to show that ReLU networks are almost always asymptotically overconfident in the sense that if one moves to infinity the confidence of the classifier approaches 1 (instead of converging to the desirable $1/K$ as in these regions the classifier has never seen any data). The following theorem now shows that, in contrast to standard ReLU networks, our proposed joint classifier gets provably less confident in its decisions as one moves away from the training data which is a desirable property of any reasonable classifier.

**Theorem 1.** *Let $x \in \mathbb{R}^d$ with $x \neq 0$ and let $g : \mathbb{R}^d \to \mathbb{R}$ be the ReLU-network of the binary discriminator (with the last activation being a non-leaky ReLU). Denote by $\{Q_r\}_{r=1}^R$ the finite set of polytopes on which $g$ is affine (exists by Lemma 1 in App. C). Denote by $Q_t$ the polytope such that $\beta x \in Q_t$ for all $\beta \geq \alpha$ and let $x^{(L-1)}(z) = Uz + d$ with $U \in \mathbb{R}^{n_{L-1} \times d}$ and $d \in \mathbb{R}^{n_{L-1}}$ be the output of the pre-logit layer of $g$ for $z \in Q_t$. If $Ux \neq 0$, then $\lim_{\beta \to \infty} \hat{p}(y|\beta x) = \frac{1}{K}$.*

The proof is in App. C. In App. A we show that the condition $Ux \neq 0$ is not restrictive, as this property holds in all cases where we checked it. The negativity condition on the weights $W_g^{(L_g)}$ of the output layer of the in-vs. out-distribution discriminator $g$ is crucial for the proof. This may seem restrictive, but we did not encounter any negative influence of this constraint on test accuracy, guaranteed or standard OOD detection performance. Thus, the asymptotic guarantees come essentially for free.

## 4   Experiments

### 4.1   Training of ProoD

We provide experiments on CIFAR10, CIFAR100 [29] and Restricted Imagenet (R.ImgNet) [53]. The latter consists of ImageNet images (ILSVRC2012) [16, 49] belonging to 9 types of animals.

**Training the Binary Discriminator** We train the binary discriminator between in-and out-distribution using the loss in Eq. (5) with the bounds over an $l_\infty$-ball of radius $\epsilon = 0.01$ for the out-distribution following [8]. We use relatively shallow CNNs with only 5 layers plus pooling layers, see App. D. For the training out-distribution, we could follow previous work and use 80M Tiny Images [52] for CIFAR10/100. However, there have been concerns over the use of this dataset [6] due to offensive class labels. Although we do not use any of the class labels, we choose to use OpenImages [30] as training OOD instead. In order to ensure a fair comparison with prior work we also present results that were obtained using 80M Tiny Images in App. E. For R.ImgNet we use the ILSVRC2012 train images that do not belong to R.ImgNet as training out-distribution (NotR.ImgNet).

**Semi-Joint Training** For the classifier we use a ResNet18 architecture on CIFAR and a ResNet50 on R.ImgNet. Note that the architecture of our binary discriminator is over an order of magnitude smaller than the one in [8] (11MB instead of 135MB) and thus the memory overhead for the binary discriminator is less than a third of that of the classifier. All schedules, hardware and hyperparameters are described in App. D. As discussed in Section 2.1, when training the binary discriminator one implicitly assumes that in- and (worst-case) out-distribution samples are equally likely. It seems very unlikely that one would be presented with such a large number of OOD samples in practice but as discussed in Section 2.1, we can adjust the weight of the losses after training the discriminator (but before training the classifier) by shifting the bias $b_g^{(L_g)}$ in the output layer of the binary discriminator. We train several ProoD models for binary shifts in $\{0, 1, 2, 3, 4, 5, 6\}$ and then evaluate the AUC and guaranteed AUC (see 4.2) on a subset of the training out-distribution OpenImages (resp. NotR.ImgNet). For all bias shifts we use the same fixed provably trained binary discriminator and only train the classifier part. As our goal is to have provable guarantees with minimal or no loss on the standard OOD detection task, among all solutions which have better AUC than outlier exposure (OE) [22] we choose the one with the highest guaranteed AUC on OpenImages (on CIFAR10/CIFAR100) resp. NotR.ImgNet (on R.ImgNet). If none of the solutions has better AUC than OE on the training out-distribution we take the one with the highest AUC. We show the trade-off curves for the example in App. D.

## 4.2 Evaluation

**Setup** For OOD evaluation for CIFAR10/100 we use the test sets from CIFAR100/10, SVHN [43], the classroom category of downscaled LSUN [55] (LSUN_CR) as well as smooth noise as suggested in [20] and described in App. D. For R.ImgNet we use Flowers [45], FGVC Aircraft [38], Stanford Cars [26] and smooth noise as test out-distributions. Since the computation of adversarial AUCs (next paragraph) requires computationally expensive adversarial attacks, we restrict the evaluation on the out-distribution to a fixed subset of 1000 images (300 in the case of LSUN_CR) for the CIFAR experiments and 400 for the R.ImgNet models. We still use the entire test set for the in-distribution. We also show the results on additional test out-distributions in App. G.

**Guaranteed and Adversarial AUC** We use the confidence of the classifier as the feature to discriminate between in- and out-distribution samples. While in standard OOD detection one uses the area under the receiver-operator characteristic (AUC) to measure discrimination of in- from out-distribution, several prior works also study the worst-case AUC (WCAUC) [41, 4, 8, 9, 5], which is defined as the minimal AUC one can achieve if each out-distribution sample is allowed to be perturbed to reach maximal confidence within a certain threat model, which in our case is an $l_\infty$-ball of radius $\epsilon$. Note that an alternative formulation of a worst-case AUC as the worst-case across all samples from the out-distribution would turn out to be uninteresting, since it would necessarily be close to zero even if only a single sample gets assigned high-confidence, so we do not consider this notion here. Formally, the AUC and WCAUC of a feature $h : \mathbb{R}^d \to \mathbb{R}$ are defined as:

$$\text{AUC}_h(p_1, p_2) = \mathop{\mathbb{E}}_{\substack{x \sim p_1 \\ z \sim p_2}} \left[ \mathbb{1}_{h(x) > h(z)} \right], \quad \text{WCAUC}_h(p_1, p_2) = \mathop{\mathbb{E}}_{\substack{x \sim p_1 \\ z \sim p_2}} \left[ \mathbb{1}_{h(x) > \max_{\|z' - z\|_\infty \le \epsilon} h(z')} \right], \quad (12)$$

where $p_1, p_2$ are in-resp. out-distribution and the indicator function $\mathbb{1}$ returns 1 if the expression in its argument is true and 0 otherwise.

For all but one of our baselines, the OOD detecting feature $h$ is the confidence of the classifier. Since the exact evaluation of the WCAUC is computationally infeasible, we compute an upper bound and

Table 2: **OOD performance:** For all models we report accuracy on the test set of the in-distribution and AUCs, guaranteed AUCs (GAUC), adversarial AUCs (AAUC) for different test out-distributions. The radius of the $l_\infty$-ball for the adversarial manipulations of the OOD data is $\epsilon = 0.01$ for all datasets. The bias shift $\Delta$ that was used for ProoD is shown for each in-distribution. The AAUCs and GAUCs for ProoD tend to be very close, indicating remarkably tight certification bounds. Models with accuracy drop of $> 3\%$ relative to the model with highest accuracy are grayed out. Of the remaining models, we highlight the best OOD detection performance.

| In: CIFAR10 | | CIFAR100 | | | SVHN | | | LSUN_CR | | | Smooth | | |
|---|---|---|---|---|---|---|---|---|---|---|---|---|---|
| | Acc | AUC | GAUC | AAUC | AUC | GAUC | AAUC | AUC | GAUC | AAUC | AUC | GAUC | AAUC |
| Plain | **95.01** | 90.0 | 0.0 | 0.7 | 93.8 | 0.0 | 0.3 | 93.1 | 0.0 | 0.5 | 98.0 | 0.0 | 0.7 |
| OE | 94.91 | **91.1** | 0.0 | 0.9 | 97.3 | 0.0 | 0.0 | **100.0** | 0.0 | 2.7 | **99.9** | 0.0 | 1.5 |
| ATOM | 93.63 | 78.3 | 0.0 | 21.7 | 94.4 | 0.0 | 24.1 | 79.8 | 0.0 | 20.1 | 99.5 | 0.0 | **73.2** |
| ACET | 93.43 | 86.0 | 0.0 | 4.0 | 99.3 | 0.0 | 4.6 | 89.2 | 0.0 | 3.7 | 99.9 | 0.0 | 40.2 |
| GOOD$_{80}$* | 87.39 | 76.7 | 47.1 | 57.1 | 90.8 | 43.4 | 76.8 | 97.4 | 70.6 | 93.6 | 96.2 | 72.9 | 89.9 |
| GOOD$_{100}$* | 86.96 | 67.8 | 48.1 | 49.7 | 62.6 | 34.9 | 36.3 | 84.9 | 74.6 | 75.6 | 87.0 | 76.1 | 78.1 |
| ProoD-Disc | - | 62.9 | 57.1 | 57.8 | 72.6 | 65.6 | 66.4 | 78.1 | 71.5 | 72.3 | 59.2 | 49.7 | 50.4 |
| ProoD $\Delta\!=\!3$ | 94.99 | 89.8 | 46.1 | **46.8** | 98.3 | 53.3 | **54.1** | 100.0 | 58.3 | **59.7** | 99.9 | **38.2** | 38.8 |

| In: CIFAR100 | | CIFAR10 | | | SVHN | | | LSUN_CR | | | Smooth | | |
|---|---|---|---|---|---|---|---|---|---|---|---|---|---|
| | Acc | AUC | GAUC | AAUC | AUC | GAUC | AAUC | AUC | GAUC | AAUC | AUC | GAUC | AAUC |
| Plain | **77.38** | **77.7** | 0.0 | 0.4 | 81.9 | 0.0 | 0.2 | 76.4 | 0.0 | 0.3 | 86.6 | 0.0 | 0.4 |
| OE | 77.25 | 77.4 | 0.0 | 0.2 | **92.3** | 0.0 | 0.0 | **100.0** | 0.0 | 0.7 | **99.5** | 0.0 | 0.5 |
| ATOM | 68.32 | 78.3 | 0.0 | 50.3 | 91.1 | 0.0 | 67.0 | 95.9 | 0.0 | 75.6 | 98.2 | 0.0 | 80.7 |
| ACET | 73.02 | 73.0 | 0.0 | 1.4 | 97.8 | 0.0 | 0.7 | 75.8 | 0.0 | 2.6 | 99.9 | 0.0 | 12.8 |
| ProoD-Disc | - | 56.1 | 52.1 | 52.3 | 61.0 | 58.2 | 58.4 | 70.4 | 66.9 | 67.1 | 29.6 | 26.4 | 26.5 |
| ProoD $\Delta\!=\!5$ | 77.16 | 76.6 | 17.3 | **17.4** | 91.5 | **19.7** | **19.8** | 100.0 | 22.5 | **23.1** | 98.9 | **9.0** | **9.0** |

| In: R.ImgNet | | Flowers | | | FGVC | | | Cars | | | Smooth | | |
|---|---|---|---|---|---|---|---|---|---|---|---|---|---|
| | Acc | AUC | GAUC | AAUC | AUC | GAUC | AAUC | AUC | GAUC | AAUC | AUC | GAUC | AAUC |
| Plain | 96.34 | 92.3 | 0.0 | 0.5 | 92.6 | 0.0 | 0.0 | 92.7 | 0.0 | 0.1 | **98.9** | 0.0 | 8.6 |
| OE | 97.10 | **96.9** | 0.0 | 0.2 | 99.7 | 0.0 | 0.4 | **99.9** | 0.0 | 1.8 | 98.0 | 0.0 | 1.9 |
| ProoD-Disc | - | 81.5 | 76.8 | 77.3 | 92.8 | 89.3 | 89.6 | 90.7 | 86.9 | 87.3 | 81.0 | 74.0 | 74.8 |
| ProoD $\Delta\!=\!4$ | **97.25** | **96.9** | 57.5 | **58.0** | 99.8 | 67.4 | **67.9** | 99.9 | 65.7 | **66.2** | 98.6 | **52.7** | **53.5** |

*Uses different architecture of classifier, see "Baselines" in Section 4.2.

a lower bound on the WCAUC by finding $\underline{h}(z) \leq \max_{\|z'-z\|_\infty \leq \epsilon} h(z') \leq \bar{h}(z)$. We find the upper bound on the WCAUC - the adversarial AUC (AAUC) - by maximizing the confidence using an adversarial attack inside the $l_\infty$-ball (i.e. finding an $\underline{h}$). We compute a lower bound on the WCAUC - the guaranteed AUC (GAUC) - by computing upper bounds on the confidence inside the $l_\infty$-ball (i.e. $\bar{h}$) via IBP. For non-provable methods, no non-trivial upper bound $\bar{h} < \infty$ is available so their GAUCs are always 0. Note that our threat model is different from adversarial robustness on the in-distribution which neither our method nor the baselines pursue. Since practical OOD detection scenarios require the selection of a threshold, we also evaluate the false positive rate (FPR) at 95% true positive rate and show the results in App. F.

Vanishing gradients [47, 3] are a significant challenge for the evaluation of AAUCs [8] even more than in the evaluation of adversarial robustness on the in-distribution as the models are trained to be "flat" on the out-distribution. Thus we use an ensemble of variants of projected gradient descent (PGD) [37] as well as the black-box SquareAttack [1] with 5000 queries. We use APGD [13] (except on RImgNet, due to a memory leak) with 500 iterations and 5 random restarts. We also use a 200-step PGD attack with momentum of 0.9 and backtracking that starts with a step size of 0.1 which is halved every time a gradient step does not increase the confidence and gets multiplied by 1.1 otherwise. As stated above, the models are trained to be flat and thus the gradients can be exactly zero which renders gradient-based optimization impossible. Therefore it is important to use a variety of different starting points. For PGD we start from: i) a decontrasted version of the image, i.e. the point that minimizes the $l_\infty$-distance to the grey image $0.5 \cdot \vec{1}$ within the threat model, ii) 3 uniformly drawn

samples from the threat model, and iii) 3 versions of the original image perturbed by Gaussian noise with $\sigma = 10^{-4}$ and then clipped to the threat model. We always clip to $[0, 1]^d$ at each step of the attack. For all attacks and models we directly optimize the final score used for OOD detection.

**Baselines**    We compare to a normally trained baseline (Plain) and outlier exposure (OE), both trained using the same architecture and hyperparameters as the classifier in ProoD. For both ATOM and ACET we found the models' OOD detection to be much less adversarially robust than claimed in [9] (see App. E) so we retrained their models using the our architecture, threat model and training out-distribution with their original code (for CIFAR10/100). Running these adversarial training procedures on ImageNet resolution is infeasibly expensive. For GOOD we also retrain using OpenImages as training OOD dataset with the code from [8] (comparisons with their pre-trained models can be found in App. E). Since they are only available on CIFAR10, we tried to train models on CIFAR100 using their code and same hyperparameters and schedules as they used for CIFAR10. This only lead to models with accuracy below $25\%$, so we do not include these models in our evaluation. Since CCU was already shown to not provide benefits over OE on OOD data that is not very far from the in-distribution (e.g. uniform noise) [41, 8] we do not include it as baseline. We also evaluate the OOD-performance of the provable binary discriminator (ProoD-Disc) that we trained for ProoD. Note that this is not a classifier and is included only for reference. All results are in Table 2.

**Results**    ProoD achieves non-trivial GAUCs on all datasets. As was also observed in [8], this shows that the IBP guarantees not only generalize to unseen samples but even to unseen distributions. In App. I we show that they even generalize to the larger threat model $\epsilon = 8/255$. In general, the gap between our GAUCs and AAUCs is extremely small. This shows that the seemingly simple IBP bounds can be remarkably tight, as has been observed in other works [17, 23]. It also shows that there would be very little benefit in applying stronger verification techniques like [10, 24, 15] in ProoD. Similarly, it demonstrates the strengths of our attack as there provably does not exist an attack that could lower the AAUCs on our ProoD model by more than 1.4% on any of the out-distributions. The bounds are also much tighter than for GOOD, which is likely due to the fact that for GOOD the confidence is much harder to optimize during an attack because it involves maximizing the confidence in an essentially random class.

For CIFAR10, on 3 out of 4 out-distributions ProoD's GAUCs are higher than ATOM's and ACET's AAUCs, i.e. our model's *provable* adversarial robustness exceeds the SOTA methods' *empirical* adversarial robustness in these cases. Note that this is *not* due to our retraining, because the authors' pre-trained models perform even more poorly (as shown in App. E). On CIFAR100, ProoD's guarantees are weaker and ATOM produces strong AAUCs. However, we observe that training both ACET and ATOM can produce inconsistent results, i.e. sometimes almost no robustness is achieved. For the successfully trained robust ATOM model on CIFAR100 we observe drastically reduced accuracy. Due to the difficulty in attacking these models, it is not unlikely that a more sophisticated attack could produce even lower AAUCs. Combined with the fact that both ACET and ATOM rely on expensive adversarial training procedures we argue that using ProoD is preferable in practice.

On CIFAR10, we see that ProoD's GAUCs are comparable to, if slightly worse than the ones of both $GOOD_{80}$ and $GOOD_{100}$. Note that although the presented GOOD models are retrained, the same observations hold true when comparing to the pre-trained models (see App. E). However, we want to point out that ProoD achieves this while retaining both high accuracy and OOD performance, both of which are lacking for GOOD. It is also noteworthy that the GOOD models' memory footprints are over twice as large as ProoD's. Generally, for ProoD the accuracy is comparable to OE and the OOD performance is similar or marginally worse. Thus ProoD shows that it is possible to achieve certifiable adversarial robustness on the out-distribution while keeping very good prediction and OOD detection performance. Note that all methods struggle on separating CIFAR10 and CIFAR100 when using OpenImages as training OOD (as compared to 80M Tiny Images in App. E).

To the best of our knowledge with R.ImgNet we provide the first worst case OOD guarantees on high-resolution images. The GAUCs are higher than on CIFAR, indicating that meaningful certificates on higher resolution are more achievable on this task than one might expect. FGVC and Cars may seem simple to separate from the animals in R.ImgNet but this cannot be said for Flowers which are difficult to provably distinguish from images of insects on flowers.

In summary, ProoD achieves our goal of maintaining high accuracy and clean OOD detection performance while providing provably adversarially robust OOD detection. In fact, out of all the

methods that do not significantly impair the in-distribution accuracy, ProoD is the only method providing such guarantees as well while simultaneously having the highest empirical robustness. Also note that for applications where adversarial robustness on the in-distribution is desired despite the induced reduction in accuracy, one can combine our ProoD model with a robustly trained classifier. In App. K, we demonstrate that ProoD in fact improves the clean as well as the robust OOD detection performance in this setting.

**Limitations & Impact**   Like our baselines, ProoD depends on a suitable training out-distribution being available. Also, as noted throughout the paper, our method only focuses on adversarial robustness around OOD samples and does not by itself achieve adversarial robustness around the in-distribution. Furthermore, our considered threat model is $l_\infty$ and while the general architecture could also be applied to other threat models, this would require moving beyond IBP. We also show in App. F that non-trivial bounds on the adversarial FPR are beyond all current methods, including ProoD.

Since our work aims to make ML models safer and more reliable, we do not anticipate any misuse of the technology. Additionally, since our method does not rely on any adversarial training, the environmental impact of training is lower than for most of our baselines. However, there have been concerns about any use of the 80M Tiny Images dataset, which is why we include these results only in App. E for back-compatibility.

## 5   Conclusion

We have demonstrated how to combine a provably adversarially robust binary discriminator between in- and out-distribution with a standard classifier in order to simultaneously achieve high accuracy, high clean OOD detection performance as well as certified adversarially robust OOD detection. Thus, we have combined the best properties of previous work with only a small increase in total model size and only a single hyperparameter. This suggests that certifiable adversarial robustness on the out-distribution (as opposed to the in-distribution) is indeed possible without losing accuracy. We further showed how in our model simply enforcing negativity in the final weights of the discriminator fixes the problem of asymptotic overconfidence in ReLU classifiers. Training ProoD models is simple and stable and thus ProoD provides OOD guarantees that come (almost) for free.

## Acknowledgements

The authors acknowledge support from the German Federal Ministry of Education and Research (BMBF) through the Tübingen AI Center (FKZ: 01IS18039A) and from the Deutsche Forschungsgemeinschaft (DFG, German Research Foundation) under Germany's Excellence Strategy (EXCnumber 2064/1, Project number 390727645), as well as from the DFG TRR 248 (Project number389792660). The authors thank the International Max Planck Research School for Intelligent Systems (IMPRS-IS) for supporting Alexander Meinke. We also thank Maximilian Augustin for helpful advice.

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
