 

Figure 2: **Left, Asymptotic confidence:** We plot the mean confidence in the predicted in-distribution class for different models as one moves away from CIFAR100 samples along the trajectories $x + \alpha n$, where $n \in [-0.5, 0.5]^d$ and $\alpha \geq 0$. Only GOOD and ProoD converge to uniform confidence. **Right, Adversarial asymptotic confidence:** We try to find adversarial directions in which ProoD remains at a constant high confidence, as opposed to converging to low confidence. We plot the *maximum* of $\hat{p}(i|x)$ across 100 adversarially chosen directions as one moves further in these directions by factors of $\alpha$. Note that $\hat{p}(i|x) \to 0$ implies $\hat{p}(y|x) \to \frac{1}{K}$.

# A    Adversarial Asymptotic Overconfidence

According to the authors of [20], under mild conditions, we should expect to find asymptotic overconfidence in all ReLU networks and almost all directions. In order to empirically evaluate this, we take different models that were trained on CIFAR10 and evaluate their confidence on different CIFAR100 samples. For each sample $x$ we track the confidence, $\max_k \hat{p}(k|x)$, along a trajectory in a uniform noise direction $x + \alpha n$, where $n \in [-0.5, 0.5]^d$ and $\alpha \geq 0$. The mean confidence across 100 such trajectories is shown on the left side of Figure 2. Even the models that produce low confidences on the original OOD sample asymptotically converge to maximal confidence far away. The only exceptions here are GOOD and ProoD and only ProoD can guarantee that the confidence cannot converge to 1.

However, even though the architecture provably prevents arbitrarily overconfident predictions and Theorem 1 ensures that most directions will indeed converge to uniform, it is, in principle, possible to find directions where the confidence $\hat{p}(i|x)$ remains constant if the condition $Ux \neq 0$ in Theorem 1 is not satisfied. We attempted to find such directions by running the following type of attack. We start from a random point $x \in [-0.5, 0.5]^d$ that we project onto a sphere of radius 100. We now run gradient descent (for 20000 steps), maximizing $g(x)$ while projecting onto the sphere at each step (unnormalized gradients with step size 0.1 for the first 10000 steps and 0.01 for the last 10000 steps). We then increase the radius to 1000 and run an additional 20000 steps with step size 0.1. We rescale the resulting direction vector down to an $l_\infty$-ball of norm 1 and compute the confidence $\hat{p}(i|x)$ as a function of the scaling in the adversarial directions. We show the resulting scale-wise *maximum* over 100 adversarial directions in Figure 2. Note that even the worst-case over 100 adversarially found directions decays to 0 asymptotically, thus empirically confirming the practical utility of Theorem 1. Note that the value of $\hat{p}(i|x)$ converging to 0 implies that the confidence of the ProoD model $\hat{p}(y|x)$ converges to 10%.

In Figure 2 GOOD also stands out as having low confidence in all directions that we studied. This is because in all the asymptotic regions that we looked at, the pre-activations of the penultimate layer are all negative. If one moves outward and these pre-activations only get more negative in all directions far away from the data, the confidence does, in fact, remain low. Unfortunately, it also leads to gradients that are precisely zero, which is why the same attack can not be applied here. However, there is no guarantee that GOOD does not also get in some direction asymptotically overconfident.

Table 3: **Architecture:** The architectures that are used for the binary discriminators. Each convolutional layer is directly followed by a ReLU.

| CIFAR | R.ImgNet |
|---|---|
| Conv2d(3, 128) | Conv2d(3, 128) |
| Conv2d(128, 256)$_{s=2}$ | AvgPool(2) |
| Conv2d(256, 256) | Conv2d(128, 256)$_{s=2}$ |
| AvgPool(2) | AvgPool(2) |
| FC(16384, 128) | Conv2d(256, 256) |
| FC(128, 1) | AvgPool(2) |
| | FC(50176, 128) |
| | FC(128, 1) |

# B   Separate Training for ProoD

In Section 2.1 we describe semi-joint training of $\hat{p}(y|x)$. However, as pointed out in that section, it is possible to separately train a certifiable binary discriminator $\hat{p}(i|x)$ and an OOD aware classifier $\hat{p}(y|x, i)$ and to then simply combine them via Eq. (2). We call this method separate training ProoD-S and evaluate it by using an OE trained model for $\hat{p}(y|x, i)$. The two versions of the methods ProoD and ProoD-S are also described in Algorithm 1 and Algorithm 2. We show the results in Table 4, where we repeat the results for OE and ProoD for the reader's convenience. Note that OE and ProoD-S must always have the same accuracy on the in-distribution since they use the same model for classification (note that (2) preserves the ranking of $\hat{p}(y|x, i)$).

We see that the AUCs of ProoD-S are almost identical to those of OE. Without almost any loss in performance ProoD-S manages to provide non-trivial GAUCs. However, as one would expect, the semi-jointly trained ProoD provides stronger guarantees at similar clean performance. Nonetheless, this post-hoc method of adding some amount of certifiability to an existing system may be interesting in applications where retraining a deployed model from scratch is infeasible.

Table 4: **Separate training:** Addendum to Table 2 showing the AUCs, GAUCs and AAUCs of ProoD-S on all datasets. The accuracy must always be identical to that of OE and the clean AUCs are also very similar to those of OE. The guarantees are almost always strictly weaker than those provided by the semi-jointly trained ProoD.

| In: CIFAR10 | Acc | CIFAR100 AUC | GAUC | AAUC | SVHN AUC | GAUC | AAUC | LSUN_CR AUC | GAUC | AAUC | Smooth AUC | GAUC | AAUC |
|---|---|---|---|---|---|---|---|---|---|---|---|---|---|
| OE | 94.91 | **91.1** | 0.0 | 0.9 | 97.3 | 0.0 | 0.0 | **100.0** | 0.0 | 2.7 | **99.9** | 0.0 | 1.5 |
| ProoD-S $\Delta=3$ | 94.91 | 89.3 | 44.7 | 45.3 | 97.3 | 51.8 | 52.6 | **100.0** | 56.7 | 57.7 | **99.9** | 36.7 | 37.6 |
| ProoD $\Delta=3$ | **94.99** | 89.8 | **46.1** | **46.8** | **98.3** | **53.3** | **54.1** | **100.0** | **58.3** | **59.7** | **99.9** | **38.2** | **38.8** |

| In: CIFAR100 | Acc | CIFAR10 AUC | GAUC | AAUC | SVHN AUC | GAUC | AAUC | LSUN_CR AUC | GAUC | AAUC | Smooth AUC | GAUC | AAUC |
|---|---|---|---|---|---|---|---|---|---|---|---|---|---|
| OE | **77.25** | **77.4** | 0.0 | 0.2 | **92.3** | 0.0 | 0.0 | **100.0** | 0.0 | 0.7 | **99.5** | 0.0 | 0.5 |
| ProoD-S $\Delta=5$ | **77.25** | **77.4** | 17.2 | 17.3 | **92.3** | 19.5 | 19.6 | **100.0** | 22.4 | 22.6 | **99.5** | 9.0 | **9.1** |
| ProoD $\Delta=5$ | 77.16 | 76.6 | **17.3** | **17.4** | 91.5 | **19.7** | **19.8** | **100.0** | **22.5** | **23.1** | 98.9 | 9.0 | 9.0 |

| In: R.ImgNet | Acc | Flowers AUC | GAUC | AAUC | FGVC AUC | GAUC | AAUC | Cars AUC | GAUC | AAUC | Smooth AUC | GAUC | AAUC |
|---|---|---|---|---|---|---|---|---|---|---|---|---|---|
| OE | 97.10 | **96.9** | 0.0 | 0.2 | 99.7 | 0.0 | 0.4 | **99.9** | 0.0 | 1.8 | 98.0 | 0.0 | 1.9 |
| ProoD-S $\Delta=4$ | 97.10 | **96.9** | 50.1 | 50.7 | 99.7 | 59.7 | 60.6 | **99.9** | 57.9 | 58.9 | 98.0 | 40.8 | 42.3 |
| ProoD $\Delta=4$ | **97.25** | **96.9** | **57.5** | **58.0** | **99.8** | **67.4** | **67.9** | **99.9** | **65.7** | **66.2** | **98.6** | **52.7** | **53.5** |

## C   Proof of Theorem 1

The following result of [20] basically says that as one moves to infinity by upscaling a vector one eventually ends up in a polytope which extends to infinity. We use this in the proof of our Theorem.

**Lemma 1** ([20]). *Let $\{Q_r\}_{r=1}^R$ be the set of convex polytopes on which a ReLU-network $f : \mathbb{R}^d \to \mathbb{R}^K$ is an affine function, that is for every $k \in \{1, \ldots, R\}$ and $x \in Q_k$ there exists $V^k \in \mathbb{R}^{K \times d}$ and $c^k \in \mathbb{R}^K$ such that $f(x) = V^k x + c^k$. For any $x \in \mathbb{R}^d$ with $x \neq 0$ there exists $\alpha \in \mathbb{R}$ and $t \in \{1, \ldots, R\}$ such that $\beta x \in Q_t$ for all $\beta \geq \alpha$.*

**Theorem 1.** *Let $x \in \mathbb{R}^d$ with $x \neq 0$ and let $g : \mathbb{R}^d \to \mathbb{R}$ be the ReLU-network of the binary discriminator (with the last activation being a non-leaky ReLU). Denote by $\{Q_r\}_{r=1}^R$ the finite set of polytopes on which $g$ is affine (exists by Lemma 1 in App. C). Denote by $Q_t$ the polytope such that $\beta x \in Q_t$ for all $\beta \geq \alpha$ and let $x^{(L-1)}(z) = Uz + d$ with $U \in \mathbb{R}^{n_{L-1} \times d}$ and $d \in \mathbb{R}^{n_{L-1}}$ be the output of the pre-logit layer of $g$ for $z \in Q_t$. If $Ux \neq 0$, then $\lim_{\beta \to \infty} \hat{p}(y|\beta x) = \frac{1}{K}$.*

*Proof.* We note that with a similar argument as in the derivation of (4) it holds

$$\hat{p}(y|\beta x) \leq \hat{p}(i|\beta x) + \frac{1}{K}\left(1 - \hat{p}(i|\beta x)\right) = \frac{K-1}{K}\hat{p}(i|\beta x) + \frac{1}{K}. \tag{13}$$

Using Lemma 1 we know that there exist a polytope $Q_t$ such that $\beta x \in Q_t$ for all $\beta \geq \alpha$. Thus for all $\beta \geq \alpha$ it holds that $\beta x \in Q_t$ so that

$$\hat{p}(i|\beta x) = \frac{1}{1 + e^{-g(\beta x)}} = \frac{1}{1 + e^{\left\langle W_g^{(L_g)}, U\beta x + d\right\rangle + b_g^{(L_g)}}}.$$

As $x_i^{(L-1)}(x) \geq 0$ for all $x \in \mathbb{R}^d$ it has to hold $(\beta Ux + d)_i \geq 0$ for all $\beta \geq \alpha$ and $i = 1, \ldots, n_{L-1}$. This implies that $(Ux)_i \geq 0$ for all $i = 1, \ldots, n_{L-1}$ and since $Ux \neq 0$ there has to exist at least one component $i^*$ such that $(Ux)_{i^*} > 0$. Moreover, $W_g^{(L_g)}$ has strictly negative components and thus for all $\beta \geq \alpha$ it holds

$$g(\beta x) = \left\langle W_g^{(L_g)}, U\beta x + d\right\rangle + b_g^{(L_g)} = \beta\left\langle W_g^{(L_g)}, Ux\right\rangle + \left\langle W_g^{(L_g)}, d\right\rangle + b_g^{(L_g)}.$$

As $\left\langle W_g^{(L_g)}, Ux\right\rangle < 0$ we get $\lim_{\beta \to \infty} g(x) = -\infty$ and thus

$$\lim_{\beta \to \infty} \hat{p}(i|\beta x) = 0.$$

Plugging this into (13) yields the result. $\qquad\square$

## D   Experimental Details

**Datasets**   We use CIFAR10 and CIFAR100 [29] (MIT license), SVHN [43] (free for non-commercial use), LSUN [55] (no license), the ILSVRC2012 split of ImageNet [16, 49] (free for non-commercial use), FGVC-Aircraft [38] (free for non-commercial use), Stanford Cars [26] (free for non-commercial use), OpenImages v4 [30] (images have a CC BY 2.0 license), Oxford 102 Flower [45] (no license) as well as 80M Tiny Images [52] (no license given, see also App. E). For the train/test splits we use the standard splits, except on 80M Tiny Images where we treat a random but fixed subset of 1000 images in the first 1,000,000 as our test set. For all datasets that get used as a test out-distribution we use a random but fixed subset of 1000 images.

Following [8], the smooth noise that is used is generated as follows. Uniform noise is generated and then smoothed using a Gaussian filter with a width that is drawn uniformly at random in $[1, 2.5]$. Each datapoint is then shifted and scaled linearly to ensure full range in $[0, 1]$, i.e. $x' = \frac{x - \min(x)}{\max(x) - \min(x)}$.

**Binary Training**   The architecture that we use for the binary discriminator is relatively shallow (5 linear layers). The architecture is shown in Table 3. Our results are fairly robust to the exact choice of architecture and significantly larger models do not necessarily lead to better results as we show in App. H. Similarly to [56, 8], we use long training schedules, running Adam for 1000 epochs, with an initial learning rate of $1e-4$ that we decrease by a factor of 5 on epochs $500, 750$ and $850$ and with a

---

**Algorithm 1** Training of ProoD

---

**Require:** Training data on in-distribution $(X, Y)_{n=1}^{N}$ and out-distribution $(Z)_{m=1}^{M}$, untrained classifier $f_\theta$, untrained binary discriminator $g_\eta$, adversarial radius $\epsilon$, batch size $b$, number of classes $K$, bias shift $\Delta$

  **for** $t$ in discriminator training steps **do**           ▷ Train the binary discriminator
      $x \leftarrow$ sample minibatch from X
      $z \leftarrow$ sample minibatch from Z
      loss $\leftarrow 0$
      **for** $r$ in $1..b$ **do**                        ▷ Parallelized in practice
          $\underline{z}_r \leftarrow \mathrm{Clip}(z_r - \epsilon, 0, 1)$            ▷ Keeps perturbed values in box
          $\overline{z}_r \leftarrow \mathrm{Clip}(z_r + \epsilon, 0, 1)$
          $\overline{g}_r \leftarrow \mathrm{IntervalBoundPropagation}(g_\eta, \underline{z}_r, \overline{z}_r)$ ▷ An algorithm that provides upper bounds on the output of $g_\eta$
          loss $\leftarrow$ loss $+ \frac{1}{b}\log\!\left(1 + \mathrm{e}^{-g_\eta(x_r)}\right) + \frac{1}{b}\log\left(1 + \mathrm{e}^{\overline{g}_r}\right)$
      **end for**
      $\eta \leftarrow \mathrm{SGD\_Update}(\eta, \mathrm{loss}, t)$
  **end for**
  $g_\eta \leftarrow g_\eta + \Delta$                             ▷ Apply the bias shift
  **for** $t$ in classifier training steps **do**       ▷ Train the classifier using semi-joint training
      $x, y \leftarrow$ sample minibatch from X, Y
      $z \leftarrow$ sample minibatch from Z
      loss $\leftarrow 0$
      **for** $r$ in $1..b$ **do**
          $p_i \leftarrow \mathrm{SoftMax}(f_\theta(x_r), y_r) \cdot \mathrm{Sigmoid}(g_\eta(x_r)) + \frac{1}{K}\left(1 - \mathrm{Sigmoid}(g_\eta(x_r))\right)$
         $p_o \leftarrow \frac{1}{K}\left(1 - \mathrm{Sigmoid}(g_\eta(z_r))\right) + \frac{1}{K}\sum_{c=1}^{K}\mathrm{SoftMax}(f_\theta(z_r), c) \cdot \mathrm{Sigmoid}(g_\eta(z_r))$
         loss $\leftarrow$ loss $+ \frac{1}{b}p_i + \frac{1}{b}p_o$
      **end for**
      $\theta \leftarrow \mathrm{SGD\_Update}(\theta, \mathrm{loss}, t)$             ▷ Only $\theta$ gets updated - not $\eta$.
  **end for**

---

batch size of 128 from the in-distribution and 128 from the out-distribution (for R.ImgNet: 50 epochs with drops at 25, 35, 45, batch sizes 32). In order to avoid large losses we also use a simple ramp up schedule for the $\epsilon$ used in IBP and we downweight the out-distribution loss during the initial phase of training by a scalar $\kappa$. Both $\epsilon$ and $\kappa$ are increased linearly from 0 to their final values (0.01 and 1, respectively) over the first 300 epochs (for R.ImgNet over the first 25 epochs). Compared to the training of [8] which sometimes fails, we found that training of the binary discriminator is very stable and even 100 epochs on CIFAR would be sufficient, but we found that longer training lead to slightly better results. Weight decay is set to $5 \cdot 10^{-4}$, but is disabled for the weights in the final layer. As data augmentation we use AutoAugment [14] for CIFAR and simple 4 pixel crops and reflections on R.ImgNet. The strict negativity of the weights leads to a negative bias of $g$ which can cause problems at an early stage if the $b_g^{(L_g)}$ is initialized at 0 and thus we choose 3 as initialization. All binary discriminators were trained on single 2080Ti GPUs, managed on a SLURM cluster. Overall, the training of a provable discriminator takes around 16h on CIFAR and 44h on R.ImgNet (wall clock time including evaluations and logging on each epoch).

**Semi-Joint Training**    On CIFAR we train for 100 epochs using SGD with momentum of 0.9 and a learning rate of 0.1 that drops by a factor of 10 on epochs 50, 75 and 90 (on R.ImgNet 75 epochs with drops at 30 and 60). For all datasets we train using a batch size of 128 (plus 128 out-distribution samples in the case of OE). The CIFAR experiments were run on single 2080Ti GPUs. This takes about 4h20min in wall clock time. In order to fit batches of 128 in-distribution samples and 128 out-distribution samples on R.ImgNet we had to train using 4 V100 GPUs in parallel. Because of batch normalization in multi-GPU training it is important to not simply stack the batches but to interlace in- and out-distribution samples. The wall clock time was around 15h for the semi-joint training on R.ImgNet. For selecting the bias we use the procedure described in Section 4. The trade-off curves for the AUC and GAUC on CIFAR100 and R.ImgNet are given in Figure 3.

---

**Algorithm 2** Training of ProoD-S

---

**Require:** Training data on in-distribution $(X, Y)_{n=1}^{N}$ and out-distribution $(Z)_{m=1}^{M}$, untrained classifier $f_\theta$, untrained binary discriminator $g_\eta$, adversarial radius $\epsilon$, batch size $b$, number of classes $K$, bias shift $\Delta$

  **for** $t$ in discriminator training steps **do**                           ▷ Train the binary discriminator

       $x \leftarrow$ sample minibatch from X

       $z \leftarrow$ sample minibatch from Z

       loss $\leftarrow 0$

       **for** $r$ in $1..b$ **do**                                      ▷ Parallelized in practice

          $\underline{z}_r \leftarrow \mathrm{Clip}(z_r - \epsilon, 0, 1)$                      ▷ Keeps perturbed values in box

          $\overline{z}_r \leftarrow \mathrm{Clip}(z_r + \epsilon, 0, 1)$

          $\overline{g}_r \leftarrow \mathrm{IntervalBoundPropagation}(g_\eta, \underline{z}_r, \overline{z}_r)$ ▷ An algorithm that provides upper bounds on the output of $g_\eta$

          loss $\leftarrow$ loss $+ \frac{1}{b} \log\left(1 + e^{-g_\eta(x_r)}\right) + \frac{1}{b} \log\left(1 + e^{\overline{g}_r}\right)$

       **end for**

       $\eta \leftarrow \mathrm{SGD\_Update}(\eta, \text{loss}, t)$

  **end for**

  $g_\eta \leftarrow g_\eta + \Delta$                                            ▷ Apply the bias shift

  **for** $t$ in classifier training steps **do**                  ▷ Train the classifier completely separately

       $x, y \leftarrow$ sample minibatch from X, Y

       $z \leftarrow$ sample minibatch from Z

       loss $\leftarrow 0$

       **for** $r$ in $1..b$ **do**

          $p_i \leftarrow \mathrm{SoftMax}(f_\theta(x_r), y_r)$                   ▷ The difference to Alg. 1 is here.

          $p_o \leftarrow \frac{1}{K} \sum_{c=1}^{K} \mathrm{SoftMax}(f_\theta(z_r), c)$        ▷ Note that $g_\eta$ does not appear in this loss.

          loss $\leftarrow$ loss $+ \frac{1}{b} p_i + \frac{1}{b} p_o$

       **end for**

       $\theta \leftarrow \mathrm{SGD\_Update}(\theta, \text{loss}, t)$

  **end for**

---

## E  80M Tiny Images as Training Out-Distribution

The 80M Tiny Images dataset has been retracted by the authors due to concerns over offensive class labels [6]. We support the decision of the community to move away from the use of 80M Tiny Images, so we choose to train our CIFAR models using a downscaled version of OpenImages v4 [30] as a training out-distribution. However, since all prior work used this dataset, we present results on 80M Tiny Images here in order to compare ProoD's performance to prior baselines. We encourage the community to use the results in Table 2 for future comparisons.

Our results on 80M Tiny Images are shown in Table 5. Apart from the models shown in Table 2 we add here the pre-trained energy-based OOD detector EB from [34] as an additional baseline for clean OOD detection. As EB is not trained robustly, as expected EB has low AAUCs. In terms of clean OOD detection it performs similarly to OE but with worse results on the more difficult OOD detection task CIFAR10 vs CIFAR100 and vice versa. For GOOD we use the pre-trained models from [8]. For ATOM and ACET we use the pre-trained models from [9]. Note that these use the densenet architecture and were actually trained to withstand attacks in the much stronger threat model of $\epsilon = \frac{8}{255}$. In the original paper, the authors claim near perfect AAUCs on this task but we can show that this is a case of gross overestimation of robustness. Even on the easier threat model $\epsilon = 0.01$ that we test in Table 5, their robustness is almost non-existent for both ATOM models and the CIFAR100 ACET model. The fact that their adversarial attacks were unable to find these samples clearly demonstrates that evaluating models adversarially is very difficult and potentially unreliable. Because of this we believe that our guarantees are a valuable contribution to the community.

## F  False Positive Rates

Since in a practical setting a threshold for OOD detection ultimately has to be chosen, it can be interesting to study the false positive rate at a fixed threshold. It is relatively standard to pick the

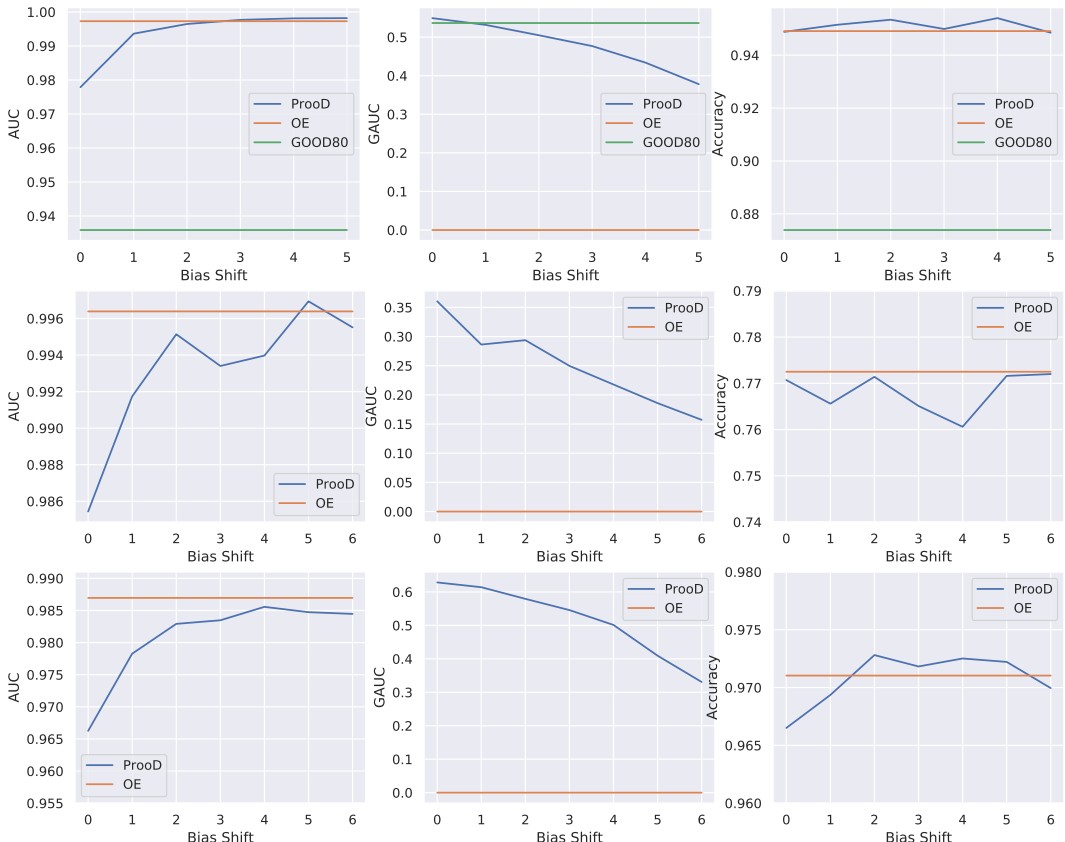

Figure 3: **Bias selection for CIFAR100 and RImgNet:** Using CIFAR10 (top), CIFAR100 (middle) and R.ImgNet (bottom) as the in-distribution and the test set of OpenImages (or NotR.ImgNet respectively) as OOD we plot the test accuracy, AUC and GAUC as a function of the bias shift $\Delta$ (see Eq. (9)).

false positive rate at 95% true positive rate (called FPR in Table 6), where low values are desirable. We show the results for all methods and datasets in Table 6. Although ProoD has similarly good performance as OE on this task, it fails to give non-trivial guarantees. Therefore achieving stronger bounds on the worst-case FPR is an interesting task for future work.

## G   Additional Datasets

In addition to the results shown in Table 2, it is interesting to study how ProoD performs on additional datasets as well as the test set of the out-distribution it was trained on. For the CIFAR datasets, we report LSUN crops, LSUN_resize, Places365 [57], iSUN [54], Textures [11], 80M Tiny Images and uniform noise in Table 7. On RImgNet we report uniform noise and the training out-distribution in Table 8.

As in Table 2 the clean performance of ProoD is comparable to that of OE, but it achieves non-trivial GAUC. On CIFAR10, $GOOD_{100}$ achieves almost perfect GAUC against uniform noise, which comes at the price of significantly worse clean AUCs on all other out-distributions, see Table 2. Almost all methods achieve very high AAUCs on uniform noise, but it is not clear if a sufficiently powerful attack could lower those scores significantly. The unusually large gap between the GAUC and AAUC of ProoD would seem to indicate that that might be the case. Surprisingly, the robustness characteristics of both ATOM and ACET vary wildly between the different datasets, sometimes appearing to be perfectly robust and on other datasets displaying no robustness at all. Especially surprising are the relatively low AAUCs on the test set of the training out-distribution.

Table 5: **Training with 80M Tiny Images:** We repeat the evaluation from Table 2 for models that were trained using 80M Tiny Images as out-distribution instead of OpenImages. Plain is identical to before and is just repeated for the reader's convenience. Note that the conclusions from the main paper still hold, which indicates that our method is robust to changes in the choice of training out-distribution. For ATOM and ACET we compare to pre-trained models from [9]. Note that these models show almost no robustness on CIFAR100 - despite the far stronger claims in [9]. Models with accuracy drop of $> 3\%$ relative to the model with highest accuracy are grayed out. Of the remaining models, we highlight the best OOD detection performance.

| In: CIFAR10 | Acc | CIFAR100 | | | SVHN | | | LSUN_CR | | | Smooth | | |
|---|---|---|---|---|---|---|---|---|---|---|---|---|---|
| | | AUC | GAUC | AAUC | AUC | GAUC | AAUC | AUC | GAUC | AAUC | AUC | GAUC | AAUC |
| Plain | 95.01 | 90.0 | 0.0 | 0.6 | 93.8 | 0.0 | 0.1 | 93.1 | 0.0 | 0.5 | 98.2 | 0.0 | 0.6 |
| OE | **95.53** | **96.1** | 0.0 | 6.0 | 99.2 | 0.0 | 0.4 | 99.5 | 0.0 | 15.2 | 99.0 | 0.0 | 11.3 |
| EB$^\star$ | 95.22 | 93.8 | 0.0 | 2.8 | 99.3 | 0.0 | 0.0 | 99.5 | 0.0 | 6.0 | 99.4 | 0.0 | 3.5 |
| ATOM$^\dagger$ | 95.20 | 93.7 | 0.0 | 14.4 | **99.6** | 0.0 | 8.6 | **99.7** | 0.0 | 40.0 | 99.6 | 0.0 | 18.8 |
| ACET$^\dagger$ | 91.48 | 91.2 | 0.0 | 80.5 | 95.3 | 0.0 | 87.6 | 98.9 | 0.0 | 95.0 | 99.9 | 0.0 | 98.3 |
| GOOD$_{80}$* | 90.13 | 87.2 | 42.5 | 63.9 | 94.2 | 37.5 | 67.4 | 93.3 | 55.2 | 83.6 | 95.3 | 57.3 | 88.5 |
| GOOD$_{100}$* | 90.14 | 70.7 | 54.5 | 55.0 | 74.9 | 56.3 | 56.6 | 75.2 | 61.0 | 61.6 | 81.4 | 66.6 | 67.5 |
| ProoD-Disc | - | 67.4 | 61.0 | 61.7 | 73.2 | 65.5 | 66.4 | 78.0 | 72.2 | 72.7 | 82.3 | 71.5 | 72.9 |
| ProoD $\Delta\!=\!3$ | 95.47 | 96.0 | **41.9** | **43.9** | 99.5 | **48.8** | **49.4** | 99.6 | **47.6** | **53.1** | **99.7** | **55.8** | **57.0** |

| In: CIFAR100 | Acc | CIFAR10 | | | SVHN | | | LSUN_CR | | | Smooth | | |
|---|---|---|---|---|---|---|---|---|---|---|---|---|---|
| | | AUC | GAUC | AAUC | AUC | GAUC | AAUC | AUC | GAUC | AAUC | AUC | GAUC | AAUC |
| Plain | **77.38** | 77.7 | 0.0 | 0.3 | 81.9 | 0.0 | 0.2 | 76.4 | 0.0 | 0.3 | 88.8 | 0.0 | 0.5 |
| OE | 77.28 | **83.9** | 0.0 | 0.8 | 92.8 | 0.0 | 0.1 | **97.4** | 0.0 | 4.6 | 97.6 | 0.0 | 0.9 |
| EB$^\star$ | 75.70 | 77.4 | 0.0 | 0.8 | **96.5** | 0.0 | 0.0 | 96.7 | 0.0 | 5.9 | **98.9** | 0.0 | 4.3 |
| ATOM$^\dagger$ | 75.06 | 64.3 | 0.0 | 0.2 | 93.6 | 0.0 | 0.2 | 97.5 | 0.0 | 9.3 | 98.5 | 0.0 | 15.0 |
| ACET$^\dagger$ | 74.43 | 79.8 | 0.0 | 0.2 | 90.2 | 0.0 | 0.0 | 96.0 | 0.0 | 2.1 | 92.9 | 0.0 | 0.3 |
| ProoD-Disc | - | 53.8 | 50.3 | 50.4 | 73.1 | 69.8 | 69.9 | 68.1 | 63.8 | 64.0 | 67.2 | 63.8 | 63.9 |
| ProoD $\Delta\!=\!1$ | 76.79 | 80.5 | **23.1** | **23.2** | 93.7 | **33.9** | **34.0** | 97.2 | **29.6** | **30.4** | 98.9 | **29.7** | **31.3** |

$^\star$ Pre-trained WideResnet from [34].
$^\dagger$Densenet architecture, using models from [9] pre-trained with $\epsilon = \frac{8}{255}$.
*CNN architecture using pre-trained models from [8].

# H   Size Ablation for Binary Discriminator

Since larger models should typically lead to better performance, we investigated the impact that model size has on the performance of our binary discriminator. We retrained ProoD-Disc models with CIFAR10 as in-distribution and 80M Tiny images as the out-distribution (since our dataloader is faster than for OpenImages which speeds up training). Since longer schedules only slightly improve results we also used shorter schedules with only 300 epochs where $\epsilon$ and $\kappa$ linearly increase from 0 to 0.01 and 1.0 respectively within the first 100 epochs and the learning rate drops occur at 150, 200 and 250 epochs. As architectures we use 10 different CNNs with different widths and depths ranging from 5 to 8 layers (for the precise architectures please refer to sizes {S, XL_b, XS, SR, SR2, C1, C3s, C3, C2, C4} in the file `provable_classifiers.py` of the code provided at `https://github.com/AlexMeinke/Provable-OOD-Detection`). We present scatter plots of the models' their performance on CIFAR100 and 80M Tiny Images in Figure 4 against their size. Clearly, there is no correlation between model size and performance and most differences are rather small, justifying our choice of a fairly small architecture in the main paper.

# I   Generalization to Larger Threat Model

Since $\epsilon = 0.01$ is a relatively weak threat model we evaluate if ProoD's guarantees actually generalize to the much stronger $\epsilon = \frac{8}{255} \approx 0.031$ that is standard in much of the literature on adversarial robustness. We use the exact same CIFAR models from Table 2 and show the results of our evaluation at $\epsilon = \frac{8}{255}$ in Table 9.

Table 6: **False positive rates:** For all models we report accuracy on the test of the in-distribution and the false positive rate at 95% true positive rate (FPR) (smaller is better). We also show the adversarial FPR (AFPR) and the guaranteed FPR (GFPR) for different test out-distributions. The radius of the $l_\infty$-ball for the adversarial manipulations of the OOD data is $\epsilon = 0.01$ for all datasets. The bias shift $\Delta$ that was used for ProoD is shown for each in-distribution. ProoD struggles to give non-trivial guarantees for the FPR@95% on most datasets. However, different from GOOD or ProoD-Disc, the clean performance is generally as good as that of OE. Models with accuracy drop of $> 3\%$ relative to the model with highest accuracy are grayed out. Of the remaining models, we highlight the best OOD detection performance.

| In: CIFAR10 | Acc | CIFAR100 | | | SVHN | | | LSUN_CR | | | Smooth | | |
|---|---|---|---|---|---|---|---|---|---|---|---|---|---|
| | | FPR | GFPR | AFPR | FPR | GFPR | AFPR | FPR | GFPR | AFPR | FPR | GFPR | AFPR |
| Plain | **95.01** | 56.3 | 100.0 | 100.0 | 40.7 | 100.0 | 100.0 | 46.7 | 100.0 | 100.0 | 10.6 | 100.0 | 100.0 |
| OE | 94.91 | 52.2 | 100.0 | 99.9 | 15.4 | 100.0 | 100.0 | **0.0** | 100.0 | 99.0 | **0.0** | 100.0 | 99.0 |
| ATOM | 93.63 | 73.4 | 100.0 | **98.8** | 33.9 | 100.0 | 100.0 | 85.3 | 100.0 | 100.0 | **0.0** | 100.0 | **86.1** |
| ACET | 93.43 | 65.4 | 100.0 | 99.5 | **3.0** | 100.0 | **99.8** | 62.7 | 100.0 | 100.0 | **0.0** | 100.0 | 89.0 |
| GOOD$_{80}$ | 87.39 | 65.1 | 100.0 | 84.8 | 26.8 | 100.0 | 48.8 | 6.0 | 100.0 | 24.7 | 19.6 | 100.0 | 52.8 |
| GOOD$_{100}$ | 86.96 | 84.8 | 100.0 | 99.3 | 87.9 | 100.0 | 99.7 | 66.0 | 100.0 | 99.0 | 68.6 | 100.0 | 98.2 |
| ProoD-Disc | - | 83.9 | 87.5 | 87.2 | 76.9 | 84.9 | 84.2 | 76.7 | 85.3 | 84.7 | 96.6 | 98.4 | 98.4 |
| ProoD $\Delta = 3$ | 94.99 | **48.0** | **99.9** | 99.7 | 9.1 | 100.0 | 100.0 | **0.0** | 100.0 | **98.7** | **0.0** | 100.0 | 100.0 |

| In: CIFAR100 | Acc | CIFAR10 | | | SVHN | | | LSUN_CR | | | Smooth | | |
|---|---|---|---|---|---|---|---|---|---|---|---|---|---|
| | | FPR | GFPR | AFPR | FPR | GFPR | AFPR | FPR | GFPR | AFPR | FPR | GFPR | AFPR |
| Plain | **77.38** | 80.1 | 100.0 | 100.0 | 77.3 | 100.0 | 100.0 | 79.0 | 100.0 | 100.0 | 70.0 | 100.0 | 100.0 |
| OE | 77.25 | 81.8 | 100.0 | 100.0 | 37.7 | 100.0 | 100.0 | 0.0 | 100.0 | 99.7 | **0.0** | 100.0 | 100.0 |
| ATOM | 68.32 | 81.3 | 100.0 | 99.6 | 51.0 | 100.0 | 97.4 | 26.3 | 100.0 | 94.7 | 10.0 | 100.0 | 86.0 |
| ACET | 73.02 | 87.9 | 100.0 | 100.0 | 8.2 | 100.0 | 100.0 | 87.0 | 100.0 | 100.0 | 0.0 | 100.0 | 98.2 |
| ProoD-Disc | - | 95.9 | 97.5 | 97.5 | 91.3 | 92.8 | 92.7 | 91.7 | 95.3 | 95.0 | 100.0 | 100.0 | 100.0 |
| ProoD $\Delta = 5$ | 77.16 | **82.2** | 100.0 | 100.0 | **37.6** | 100.0 | 100.0 | **0.0** | 100.0 | **99.3** | 3.4 | 100.0 | 100.0 |

| In: R.ImgNet | Acc | Flowers | | | FGVC | | | Cars | | | Smooth | | |
|---|---|---|---|---|---|---|---|---|---|---|---|---|---|
| | | FPR | GFPR | AFPR | FPR | GFPR | AFPR | FPR | GFPR | AFPR | FPR | GFPR | AFPR |
| Plain | 96.34 | 55.2 | 100.0 | 100.0 | 48.2 | 100.0 | 100.0 | 75.2 | 100.0 | 100.0 | **0.0** | 100.0 | 100.0 |
| OE | 97.10 | **18.2** | 100.0 | 100.0 | **0.2** | 100.0 | 100.0 | **0.0** | 100.0 | 100.0 | **0.0** | 100.0 | 100.0 |
| ProoD-Disc | - | 59.2 | 65.2 | 65.0 | 51.0 | 67.8 | 66.5 | 51.7 | 63.7 | 62.3 | 100.0 | 100.0 | 100.0 |
| ProoD $\Delta = 4$ | **97.25** | 18.5 | 100.0 | 100.0 | 0.5 | 100.0 | 100.0 | **0.0** | 100.0 | 100.0 | **0.0** | 100.0 | 100.0 |

Perhaps surprisingly, ProoD's guarantees generalize remarkably well to the much larger radius on CIFAR10. The same holds for GOOD, as was already observed in [8]. ATOM and ACET are not robust at this radius, despite having lower accuracy, lower clean OOD performance and more expensive training. On the other hand ProoD's guarantees on CIFAR100 are quite weak at this radius, but given ATOM's and ACET's low AAUCs here and the fact that GOOD cannot be trained at all on CIFAR100, the results are not worse than for the competitors.

## J   Error Bars

In order to be mindful of our resource consumption we restrict the computation of error bars to our experiments on CIFAR10. Additionally, because the dataloader was much faster we ran these experiments using 80M Tiny Images as an out-distribution as opposed to OpenImages. We reran our experiments using the same hyperparameters 5 times. We computed the mean and the standard deviations for our models for all metrics shown in Table 5. The results are shown in Table 10. We see that the fluctuations across different runs are indeed rather small. Furthermore, the clean performance of OE and ProoD show no significant discrepancies.

Table 7: **Additional datasets:** We show the AUC, AAUC and GAUC for all models on uniform noise and on the test set of the train out-distribution. Models with accuracy drop of $> 3\%$ relative to the model with highest accuracy are grayed out. Of the remaining models, we highlight the best OOD detection performance.

| In: CIFAR10 | Acc | LSUN AUC | LSUN GAUC | LSUN AAUC | LSUN_resize AUC | LSUN_resize GAUC | LSUN_resize AAUC | Places365 AUC | Places365 GAUC | Places365 AAUC | iSUN AUC | iSUN GAUC | iSUN AAUC |
|---|---|---|---|---|---|---|---|---|---|---|---|---|---|
| Plain | **95.01** | 95.9 | 0.0 | 1.3 | 95.0 | 0.0 | 8.7 | 89.5 | 0.0 | 0.3 | 94.5 | 0.0 | 10.4 |
| OE | 94.91 | 98.7 | 0.0 | 0.9 | 97.4 | 0.0 | 6.1 | **99.9** | 0.0 | 2.5 | 97.6 | 0.0 | 9.4 |
| ATOM | 93.63 | 77.3 | 0.0 | 12.1 | **100.0** | 0.0 | **98.4** | 82.6 | 0.0 | 22.0 | **100.0** | 0.0 | **98.8** |
| ACET | 93.43 | 89.2 | 0.0 | 2.5 | **100.0** | 0.0 | 91.3 | 88.0 | 0.0 | 3.4 | **100.0** | 0.0 | 92.1 |
| GOOD80 | 87.39 | 96.5 | 68.4 | 89.7 | 87.4 | 61.7 | 69.5 | 96.8 | 58.8 | 90.2 | 87.1 | 58.9 | 70.3 |
| GOOD100 | 86.96 | 95.0 | 86.0 | 86.7 | 81.1 | 67.6 | 67.9 | 74.4 | 59.7 | 60.9 | 77.5 | 63.3 | 65.2 |
| ProoD-Disc | - | 95.8 | 94.1 | 94.2 | 76.4 | 70.3 | 71.5 | 76.6 | 71.1 | 71.5 | 74.9 | 69.0 | 70.3 |
| ProoD $\Delta = 3$ | 94.99 | **99.2** | **82.2** | **82.3** | 97.1 | **57.7** | 59.2 | **99.9** | **58.7** | **59.6** | 97.1 | **56.4** | 58.1 |

| In: CIFAR10 | Acc | Uniform AUC | Uniform GAUC | Uniform AAUC | Textures AUC | Textures GAUC | Textures AAUC | 80M Tiny Images AUC | 80M Tiny Images GAUC | 80M Tiny Images AAUC | OpenImages AUC | OpenImages GAUC | OpenImages AAUC |
|---|---|---|---|---|---|---|---|---|---|---|---|---|---|
| Plain | **95.01** | 97.9 | 0.0 | 83.1 | 92.0 | 0.0 | 8.6 | 91.1 | 0.0 | 0.8 | 84.0 | 0.0 | 0.4 |
| OE | 94.91 | 99.9 | 0.0 | 98.2 | **99.9** | 0.0 | 13.8 | 93.6 | 0.0 | 0.6 | 99.7 | 0.0 | 2.5 |
| ATOM | 93.63 | **100.0** | 0.0 | **100.0** | 97.5 | 0.0 | **74.7** | 82.8 | 0.0 | 26.7 | 73.4 | 0.0 | 22.3 |
| ACET | 93.43 | **100.0** | 0.0 | 99.9 | 98.0 | 0.0 | 41.6 | 89.0 | 0.0 | 7.8 | 81.7 | 0.0 | 5.6 |
| GOOD80 | 87.39 | 99.1 | 83.9 | 98.1 | 96.1 | 54.7 | 89.9 | 84.7 | 50.5 | 66.8 | 93.6 | 53.7 | 85.6 |
| GOOD100 | 86.96 | 94.6 | 86.1 | 89.3 | 71.0 | 49.9 | 57.1 | 74.5 | 56.7 | 58.2 | 69.5 | 54.2 | 55.6 |
| ProoD-Disc | - | 53.8 | 46.7 | 46.7 | 75.3 | 69.9 | 70.5 | 75.7 | 69.8 | 70.9 | 64.8 | 59.0 | 59.6 |
| ProoD $\Delta = 3$ | 94.99 | 99.9 | 35.0 | 94.7 | **99.9** | **57.6** | 63.1 | **93.8** | **57.5** | **59.0** | 99.8 | **47.7** | **49.7** |

| In: CIFAR100 | Acc | LSUN AUC | LSUN GAUC | LSUN AAUC | LSUN_resize AUC | LSUN_resize GAUC | LSUN_resize AAUC | Places365 AUC | Places365 GAUC | Places365 AAUC | iSUN AUC | iSUN GAUC | iSUN AAUC |
|---|---|---|---|---|---|---|---|---|---|---|---|---|---|
| Plain | **77.38** | 84.5 | 0.0 | 0.9 | 78.9 | 0.0 | 3.9 | 75.7 | 0.0 | 0.6 | 79.4 | 0.0 | 5.6 |
| OE | 77.25 | 94.5 | 0.0 | 1.3 | **88.2** | 0.0 | 2.1 | **99.7** | 0.0 | 1.9 | **88.3** | 0.0 | 3.1 |
| ATOM | 68.32 | 83.4 | 0.0 | 66.7 | 92.1 | 0.0 | 71.3 | 87.0 | 0.0 | 62.1 | 91.7 | 0.0 | 73.7 |
| ACET | 73.02 | 81.3 | 0.0 | 4.3 | 100.0 | 0.0 | 77.0 | 74.8 | 0.0 | 4.3 | 100.0 | 0.0 | 78.2 |
| ProoD-Disc | - | 81.4 | 78.8 | 79.3 | 70.1 | 66.7 | 67.1 | 71.7 | 68.2 | 68.5 | 69.3 | 65.6 | 66.0 |
| ProoD $\Delta = 5$ | 77.16 | **94.8** | **28.5** | **28.7** | 86.3 | **22.6** | **22.8** | **99.7** | **23.6** | **23.8** | 87.2 | **22.1** | **22.4** |

| In: CIFAR100 | Acc | Uniform AUC | Uniform GAUC | Uniform AAUC | Textures AUC | Textures GAUC | Textures AAUC | 80M Tiny Images AUC | 80M Tiny Images GAUC | 80M Tiny Images AAUC | OpenImages AUC | OpenImages GAUC | OpenImages AAUC |
|---|---|---|---|---|---|---|---|---|---|---|---|---|---|
| Plain | **77.38** | 82.7 | 0.0 | 53.0 | 77.5 | 0.0 | 5.3 | 79.7 | 0.0 | 1.2 | 75.5 | 0.0 | 0.7 |
| OE | 77.25 | 99.3 | 0.0 | **91.4** | 99.0 | 0.0 | 7.5 | 80.2 | 0.0 | 1.5 | 99.6 | 0.0 | 1.0 |
| ATOM | 68.32 | 100.0 | 0.0 | 100.0 | 90.0 | 0.0 | 66.6 | 90.1 | 0.0 | 70.0 | 83.7 | 0.0 | 58.9 |
| ACET | 73.02 | 100.0 | 0.0 | 99.9 | 91.5 | 0.0 | 21.0 | 77.0 | 0.0 | 6.8 | 74.5 | 0.0 | 3.0 |
| ProoD-Disc | - | 40.9 | 37.0 | 37.4 | 53.6 | 50.2 | 50.3 | 59.7 | 56.7 | 56.9 | 58.3 | 54.6 | 54.8 |
| ProoD $\Delta = 5$ | 77.16 | **99.7** | **12.1** | 87.0 | **99.1** | **16.8** | **22.6** | **80.5** | **19.5** | **19.7** | **99.7** | **18.6** | **20.1** |

# K   Combining ProoD with a Robust Classifier

In this work we have provided guarantees on adversarially robust out-of-distribution detection that do not come at the cost of accuracy. This is the main reason why we did not consider adversarial robustness on the in-distribution since this is known to come at the cost of clean accuracy [53]. However, it is an interesting question if ProoD can nonetheless be applied to models that are also adversarially robust on the in-distribution. In order to illustrate that it in fact can, we combine an adversarially robust Resnet-18 [18] from RobustBench [12] with a robust accuracy of 58.5% at $l_\infty$ $\epsilon = 8/255$ on CIFAR10 with our binary discriminator as described in App. B. We use the same bias shift of $\Delta = 3$ as for our ProoD model on CIFAR10. We call the robust model "Robust" and the combined model "Robust-ProoD". Note that because no retraining is necessary, both clean accuracy and robust accuracy of both models are guaranteed to stay the same (ProoD does not change the

Table 8: **Additional datasets:** For RImgNet, we show the AUC, AAUC and GAUC for all models on uniform noise and on the test set of the train out-distribution, i.e. NotRImgNet.

| In: R.ImgNet | Acc | Uniform | | | NotR.ImgNet | | |
|---|---|---|---|---|---|---|---|
| | | AUC | GAUC | AAUC | AUC | GAUC | AAUC |
| Plain | 96.34 | 99.3 | 0.0 | 74.9 | 91.7 | 0.0 | 0.2 |
| OE | 97.10 | 99.6 | 0.0 | 84.6 | **98.7** | 0.0 | 1.2 |
| ProoD-Disc | - | 99.7 | 99.2 | 99.3 | 73.6 | 69.9 | 69.9 |
| ProoD $\Delta = 4$ | **97.25** | **99.8** | **79.7** | **95.2** | 98.6 | **50.1** | **51.3** |

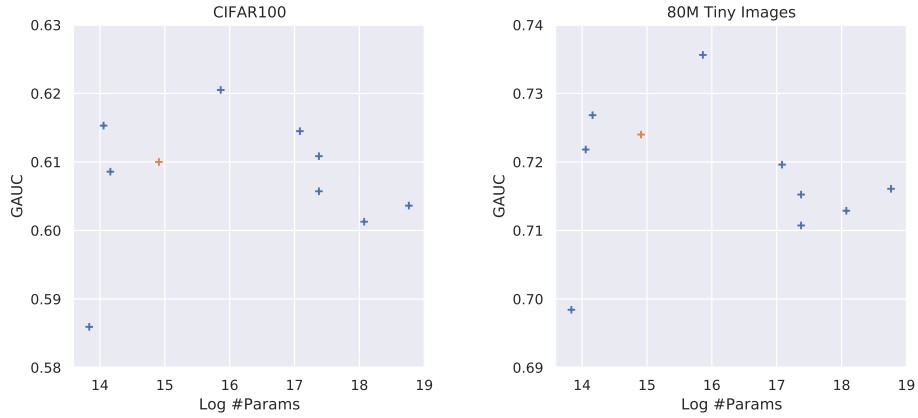

Figure 4: **Bigger models do not yield better guarantees:** We show scatter plots of the GAUC of different architectures against the log of the number of trainable parameters in the model. The orange cross indicates the architecture that is used in the main paper. There is no clear dependence of performance on model size, so it is preferable to use fairly small models.

predictions). We evaluate the adversarial robustness on the out-distribution and report the results in Table 11.

We can see that while the robust model already has remarkably strong empirical robustness on the out-distribution, ProoD does not harm the OOD detection performance of the model - neither clean nor adversarial. In fact, in all but a single case ProoD strictly improves the results (except for a $0.1\%$ drop in clean AUC on LSUN_CR). In addition to this, ProoD provides non-zero GAUCs as well as the guarantees on asymptotically low confidence. The fact that ProoD operates so well in a regime for which it was not designed highlights its versatility.

Table 9: **Generalization to Larger $\epsilon$:** We evaluate all CIFAR models in Table 2 using an $\epsilon = \frac{8}{255}$, and thus an unseen threat model. The provable methods GOOD and ProoD generalize surprisingly well, while neither ATOM nor ACET display any generalization to the larger threat model. l

| In: CIFAR10 | Acc | CIFAR100 | | | SVHN | | | LSUN_CR | | | Smooth | | |
|---|---|---|---|---|---|---|---|---|---|---|---|---|---|
| | | AUC | GAUC | AAUC | AUC | GAUC | AAUC | AUC | GAUC | AAUC | AUC | GAUC | AAUC |
| Plain | **95.01** | 90.0 | 0.0 | 0.0 | 93.8 | 0.0 | 0.0 | 93.1 | 0.0 | 0.0 | 98.0 | 0.0 | 0.0 |
| OE | 94.91 | **91.1** | 0.0 | 0.1 | 97.3 | 0.0 | 0.0 | **100.0** | 0.0 | 0.1 | **99.9** | 0.0 | 0.0 |
| ATOM | 93.63 | 78.3 | 0.0 | 1.3 | 94.4 | 0.0 | 1.5 | 79.8 | 0.0 | 0.2 | 99.5 | 0.0 | 9.6 |
| ACET | 93.43 | 86.0 | 0.0 | 1.1 | **99.3** | 0.0 | 1.1 | 89.2 | 0.0 | 0.8 | **99.9** | 0.0 | 3.8 |
| GOOD80* | 87.39 | 76.7 | 37.5 | 51.6 | 90.8 | 38.6 | 74.3 | 97.4 | 57.6 | 90.2 | 96.2 | 61.1 | 87.8 |
| GOOD100* | 86.96 | 67.8 | 39.4 | 43.5 | 62.6 | 29.0 | 30.9 | 84.9 | 67.6 | 70.7 | 87.0 | 63.3 | 69.2 |
| ProoD-Disc | - | 62.9 | 44.1 | 46.1 | 72.6 | 52.5 | 57.1 | 78.1 | 56.3 | 58.9 | 59.2 | 34.9 | 37.2 |
| ProoD $\Delta=3$ | 94.99 | 89.8 | **39.2** | **41.0** | 98.3 | **46.9** | **50.8** | 100.0 | **50.2** | **52.7** | 99.9 | **30.4** | **30.6** |

| In: CIFAR100 | Acc | CIFAR10 | | | SVHN | | | LSUN_CR | | | Smooth | | |
|---|---|---|---|---|---|---|---|---|---|---|---|---|---|
| | | AUC | GAUC | AAUC | AUC | GAUC | AAUC | AUC | GAUC | AAUC | AUC | GAUC | AAUC |
| Plain | **77.38** | **77.7** | 0.0 | 0.4 | 81.9 | 0.0 | 0.2 | 76.4 | 0.0 | 0.3 | 86.6 | 0.0 | 0.3 |
| OE | 77.25 | 77.4 | 0.0 | 0.2 | **92.3** | 0.0 | 0.0 | **100.0** | 0.0 | 0.7 | **99.5** | 0.0 | 0.5 |
| ATOM | 68.32 | 78.3 | 0.0 | 10.4 | 91.1 | 0.0 | 15.2 | 95.9 | 0.0 | 23.0 | 98.2 | 0.0 | 23.5 |
| ACET | 73.02 | 73.0 | 0.0 | 1.4 | 97.8 | 0.0 | 0.7 | 75.8 | 0.0 | 2.6 | 99.9 | 0.0 | 3.8 |
| ProoD-Disc | - | 56.1 | 41.1 | 43.1 | 61.0 | 50.5 | 51.8 | 70.4 | 57.5 | 58.8 | 29.6 | 20.9 | 20.8 |
| ProoD $\Delta=5$ | 76.51 | 76.6 | **13.7** | **14.1** | 91.5 | **16.9** | **16.9** | 100.0 | **18.1** | **18.2** | 98.9 | **8.1** | **8.1** |

| In: R.ImgNet | Acc | Flowers | | | FGVC | | | Cars | | | Smooth | | |
|---|---|---|---|---|---|---|---|---|---|---|---|---|---|
| | | AUC | GAUC | AAUC | AUC | GAUC | AAUC | AUC | GAUC | AAUC | AUC | GAUC | AAUC |
| Plain | 96.34 | 92.3 | 0.0 | 0.0 | 92.6 | 0.0 | 0.0 | 92.7 | 0.0 | 0.0 | **98.9** | 0.0 | 0.0 |
| OE | 97.10 | **96.9** | 0.0 | 0.2 | 99.7 | 0.0 | 0.0 | **99.9** | 0.0 | 0.0 | 98.0 | 0.0 | 0.0 |
| ProoD-Disc | - | 81.5 | 60.4 | 61.4 | 92.8 | 78.0 | 80.8 | 90.7 | 76.3 | 79.2 | 81.0 | 47.3 | 53.7 |
| ProoD $\Delta=4$ | **97.25** | **96.9** | 42.8 | 45.0 | **99.8** | 57.0 | 59.4 | 99.9 | 56.0 | 58.7 | 98.6 | **31.6** | **36.3** |

*Uses different architecture of classifier, see "Baselines" in Section 4.2.

Table 10: **Error bars:** We show the mean and standard deviation $\sigma$ of all metrics for our CIFAR10 models across 5 runs. The tolerances for ProoD's clean performance are very small and yet the differences in clean performance between OE ProoD are not significant.

| In: CIFAR10 | Acc | CIFAR100 | | | SVHN | | | LSUN_CR | | | Smooth | | |
|---|---|---|---|---|---|---|---|---|---|---|---|---|---|
| | | AUC | GAUC | AAUC | AUC | GAUC | AAUC | AUC | GAUC | AAUC | AUC | GAUC | AAUC |
| Plain | 94.91 | 90.0 | 0.0 | 0.6 | 93.9 | 0.0 | 0.1 | 93.4 | 0.0 | 0.7 | 96.7 | 0.0 | 1.2 |
| Plain $\sigma$ | 0.16 | 0.1 | 0.0 | 0.1 | 1.2 | 0.0 | 0.0 | 0.3 | 0.0 | 0.2 | 2.1 | 0.0 | 0.5 |
| OE | 95.56 | 96.1 | 0.0 | 7.6 | 99.4 | 0.0 | 0.4 | 99.6 | 0.0 | 16.7 | 99.6 | 0.0 | 4.3 |
| OE $\sigma$ | 0.04 | 0.1 | 0.0 | 1.5 | 0.1 | 0.0 | 0.2 | 0.1 | 0.0 | 3.5 | 0.3 | 0.0 | 3.7 |
| ProoD-Disc | - | 67.7 | 61.6 | 62.2 | 75.5 | 68.6 | 69.3 | 76.5 | 70.4 | 70.9 | 87.2 | 77.7 | 78.8 |
| ProoD-Disc $\sigma$ | - | 0.7 | 0.7 | 0.7 | 1.4 | 1.7 | 1.5 | 1.4 | 1.7 | 1.7 | 3.6 | 4.3 | 4.3 |
| ProoD $\Delta=3$ | 95.60 | 96.0 | 42.2 | 44.1 | 99.4 | 48.6 | 49.2 | 99.6 | 47.1 | 52.0 | 99.8 | 55.2 | 57.0 |
| ProoD $\Delta=3\ \sigma$ | 0.11 | 0.1 | 0.8 | 0.8 | 0.1 | 0.6 | 0.6 | 0.1 | 1.5 | 1.9 | 0.1 | 2.9 | 3.4 |

Table 11: **Robust models:** We report the OOD detection performance (AUCs, AAUCs and GAUCs) of a model that is adversarially robust on the in-distribution for different test out-distributions. The radius of the $l_\infty$-ball for the adversarial manipulations of the OOD data is $\epsilon = 0.01$ for all datasets.

| In: CIFAR10 | Acc | CIFAR100 | | | SVHN | | | LSUN_CR | | | Smooth | | |
|---|---|---|---|---|---|---|---|---|---|---|---|---|---|
| | | AUC | GAUC | AAUC | AUC | GAUC | AAUC | AUC | GAUC | AAUC | AUC | GAUC | AAUC |
| Robust | 87.35 | 82.7 | 0.0 | 75.3 | 90.0 | 0.0 | 84.0 | 89.7 | 0.0 | 82.9 | **94.5** | 0.0 | 85.7 |
| Robust-ProoD | 87.35 | **82.9** | **14.3** | **75.6** | **90.4** | **17.9** | **84.5** | **90.1** | **18.8** | **83.4** | 94.4 | **10.3** | **85.8** |