# OpenReview forum: "Provably Adversarially Robust Detection of Out-of-Distribution Data (Almost) for Free"
_NeurIPS.cc/2022/Conference — NeurIPS 2022 Accept_

### Official Review · Reviewer_ejdX · 2022-06-19

**Rating:** 4
**Confidence:** 5
**Soundness:** 2 fair
**Presentation:** 3 good
**Contribution:** 2 fair

**Summary:**

This paper proposes ProoD which merges a certified binary classifier for in-versus out-distribution with a classifier for the in-distribution task in a principled fashion into a joint classifier. It shows that ProoD simultaneously achieves three properties: 1) guaranteed adversarially robust OOD detection via certified upper bounds on the confidence in $l_\infty$-balls around OOD samples; 2) ProoD provably prevents the asymptotic overconfidence of deep neural networks; 3) ProoD can be used with arbitrary architectures and has no loss in prediction performance and standard OOD detection performance. Extensive experiments are performed to show the promising performance of the proposed method ProoD.

**Questions:**

1. How to interpret the experimental results? Does the proposed method ProoD achieve a good trade-off between clean accuracy and adversarial robustness? Does ProoD outperform existing methods in terms of adversarial robustness?

2. How to select a suitable threshold for the proposed method ProoD? Does ProoD have good performance under the False Positive Rate at 95% true positive rate (FPR) metric?

**Limitations:**

The limitations and potential negative societal impact are properly addressed.

**Strengths And Weaknesses:**

I think this paper has the following strengths:

1. It derives the method ProoD in a principled way and proves that the method can prevent the asymptotic overconfidence of deep neural networks. Also, ProoD can be used with arbitrary architectures and has no loss in prediction performance and standard OOD detection performance;

2. It performs extensive experiments to evaluate the proposed method ProoD and also compares ProoD to existing baselines;

3. It is well-written and the related works are properly discussed.

However, I think this paper has the following weaknesses:

1. The idea of training a discriminator independently via interval bound propagation (IBP) to achieve certified robustness on out-of-distribution samples is not very novel since IBP is an existing technique that can ensure certified robustness;

2. The performance of the proposed method ProoD is quite mixed across different in-distribution and OOD datasets. For example, in Table 2, on CIFAR-10 vs. Smooth, the performance of ProoD is worse than that of ATOM and GOOD in terms of AAUC metric, and in terms of GAUC metric, the performance of ProoD is also worse than that of GOOD; On CIFAR-10 vs. LSUN_CR, the performance of ProoD is also worse than that of GOOD in terms of GAUC metric. On CIFAR-100 as the in-distribution dataset, it seems the performance of ProoD is usually worse than that of ATOM in terms of the AAUC metric. In Table 7 (in the appendix), when evaluating the method on more OOD test datasets, the performance of ProoD is also mixed. Although ProoD usually has higher accuracy compared to the baselines, its adversarial robustness is worse when it achieves higher accuracy. So there is a trade-off here. For robust OOD detection, I think we should consider both clean accuracy and adversarial robustness. However, it seems ProoD doesn't achieve good performance when considering both clean accuracy and adversarial robustness (or achieve a good trade-off).

3. In practice, we need to pick a threshold for OOD detection. Thus, the False Positive Rate at 95% true positive rate (FPR) metric is also very important. Some results under the FPR metric are reported in Appendix F. However, based on the results in Table 6 (Appendix F), it seems the performance of ProoD is not good, and ProoD doesn't have clear advantages over the existing methods under the FPR metric. Thus, it might be hard to select a suitable threshold for the proposed method ProoD.

---

> ### Author Response · Authors · 2022-07-30
> **Response to Reviewer ejdX**
>
> **We thank the reviewer for their comments.**
>
> *“How to interpret the experimental results? Does the proposed method ProoD achieve a good trade-off between clean accuracy and adversarial robustness? Does ProoD outperform existing methods in terms of adversarial robustness?“*
>
> As we state in the paper we do not trade off any accuracy because we do not pursue any adversarial robustness on the in-distribution. If we were to adversarially train on the in-distribution then our trade-off would be no better or worse than that of any other adversarial training method. Also see our response to Reviewer 7nMZ.
>
> *“The performance of the proposed method ProoD is quite mixed across different in-distribution and OOD datasets.”*
>
> We emphasize that we intend to provide a method that can be used in practice, where a significant drop in accuracy is simply not acceptable. Out of all the models that have high accuracy, ProoD is the only one that provides any out-distribution robustness at all, and even does so provably. The only times where ProoD gets outperformed on individual metrics is by models with significantly impaired accuracy or clean OOD detection performance.
>
> *“How to select a suitable threshold for the proposed method ProoD?”*
>
> We agree with the reviewer that the problem of selecting a threshold is clearly very important in practice. In those cases, practitioners should attempt to estimate the base rates of in- and out-distribution samples as well as the relative costs of false positives to false negatives from which a suitable threshold could be determined. Since no such values are available in our setting, we do not consider this issue. Note that the same is true for most works on OOD detection including every single one of our baselines.
>
> *“Does ProoD have good performance under the False Positive Rate at 95% true positive rate (FPR) metric?”*
>
> Yes, in fact ProoD has excellent performance on the commonly used FPR@95% TPR metric compared to all other models on all datasets (as we show in Table 6 in Appendix F). We are not quite sure what makes the reviewer say that “based on the results in Table 6 (Appendix F), it seems the performance of ProoD is not good”. Note that for FPR lower numbers are better.
> If the reviewer is referring to the adversarial FPR, then note that there currently exists no model that achieves non-trivial performance on this task while retaining high accuracy. We agree that this will be an interesting challenge for future work.

---

### Official Review · Reviewer_XeAG · 2022-07-11

**Rating:** 5
**Confidence:** 4
**Soundness:** 2 fair
**Presentation:** 3 good
**Contribution:** 3 good

**Summary:**

This paper studies a significant and interesting topic: robust Out-of-Distribution (OOD) detection issue. Specifically, the authors propose a new method which combines a provably adversarially robust binary discriminator and standard classifier. Then, authors show that their method can simultaneously achieve high accuracy, high clean OOD detection performance as well as certified adversarially robust OOD detection. Finally, the authors verify the effectiveness of their method through extensive experiments.

**Questions:**

see the weaknesses above.

**Strengths And Weaknesses:**

Strengths

1，	Training Prood model proposed in this paper is easy and stable and thus Prood provides OOD guarantees that come for free.

2，	The authors provide a theoretical guarantee for their proposed method.


Weaknesses:

1，	Related work is not presented clearly. It is hard to follow the intuition or motivation of authors.

2，	I would have preferred to see more authors’ comparison with previous related work

---

> ### Author Response · Authors · 2022-07-30
> **Response to Reviewer XeAG**
>
> **We thank the reviewer for their comments.**
>
> *“Related work is not presented clearly. It is hard to follow the intuition or motivation of authors.”*
>
> Since all other reviewers agree that our paper is “well-written” (Reviewer 7nMZ, Reviewer ejdX) and “well-motivated” (Reviewer vBCS) and that “related works are properly discussed” (Reviewer ejdX), we kindly ask the reviewer to provide us with more detailed feedback as to what they would like to see improved.
>
> *“I would have preferred to see more authors’ comparison with previous related work”*
>
> As far as we are aware, there are no other previous works which achieve provable adversarial robustness on OOD data except GOOD to which we compare. We kindly ask the reviewer which works they had in mind so that we can add and compare to them.

---

### Official Review · Reviewer_vBCS · 2022-07-11

**Rating:** 6
**Confidence:** 3
**Soundness:** 3 good
**Presentation:** 2 fair
**Contribution:** 3 good

**Summary:**

The paper proposes a noval approach, ProoD, to integrate provably robustness against adversary on OOD data into OOD detection and achieve high classification accuracy. In partcular, Prood first learns a certifably robust model for the binary classification task between in- and out-distribution, and enable its integration of with state-of-the-art OOD detection methods, like OE, via joint training. Prood demonstrated its effectiveness against asymptotic overconfidence of deep neural network both theoretically and empirically. The authors also perform comprehensive empirical study for Prood on benchmark datasets and against various SOTA OOD methods.


**Questions:**

- Since binary shift is a core hyper-parameter for joint training, can authors explain why AUC increases with bias shift whereas GAUC decreases with the bias shift?
- Since the adversarial robustness is evaluated on OOD detection task instead of the naive classification, I am wondering if the authors explore the model-specific adaptive attack? It seems that currently the authors use an ensemble of variants of projected gradient descent. Particularly, I am not sure how robust the joint training procedure is against adaptive attacks.
- I am wondering empirically what is the computation cost of the different variants of Prood, like Prood-S, Prood-disc, relative to other OOD detection methods, like OE?


**Limitations:**

- As the authors mentioned in the paper, their approach only handles adversary around OOD samples instead of the in-distribution samples, and the studied adversary perturbation is generally limited to l_\infty ball.
- The type of allowed adversary, limited to small perturbation sets, could be unrealistic for OOD data, which can have more different forms of uncertainty for the adversary.

**Strengths And Weaknesses:**

Strength:
- The paper considers a novel setup of provably adversarially robust OOD, and the Prood is well-motivated with theoretical soundness.
- The guarantee on asymptotic confidence is an interesting aspect with empirical validation.
- The empirical study is comprehensive and evaluated on various benchmark datasets

Weakness:
- The proposed variation of the algorithms though well-motivated, is poorly presented. For better presentation, I suggest the author create a latex algorithm box and describe the different options: separate training, semi-joint training, Prood-disc, and Prood-\Delta. Does Prood-disc corresponds to Prood-\Delta = 0?
- The empirical results require more clarification:
	- Highlight the best performing algorithm in each setup for table 2 and tables in the appendix
	- Explain specifically how the bias shift is chosen. I am assuming the authors take the bias shift that maximizes AUC+GAUC+Accuracy. Is there results for AAUC under different bias shift?

typos:
- PDG should be PGD for line 289

---

> ### Author Response · Authors · 2022-07-30
> **Response to Reviewer vBCS**
>
> **We thank the reviewer for the helpful comments and for spotting a typo.**
>
> *“For better presentation, I suggest the author create a latex algorithm box and describe the different options”*
>
> Thanks a lot for the good suggestion for improving our paper. We have now included this in Appendix D but if space-wise feasible we put it into the main paper for the final version.
>
> *“Does Prood-disc corresponds to Prood-\Delta = 0?“*
>
> No. ProoD-Disc indicates the binary model $\hat{p}(i|x)$ without combining it with the classifier. While we do use $\Delta=0$ when evaluating it, all AUCs - clean and worst-case - would be identical if we introduced a shift in ProoD-Disc as it changes only the absolute value of $\hat{p}(i|x)$ but not the ranking. However, the shift matters when combined with the classifier.
>
> *“Explain specifically how the bias shift is chosen”*
>
> We describe this in line 242-250. To summarize, we chose the $\Delta$ from the set $\lbrace 0,1,2,3,4,5,6 \rbrace$ such that the clean AUC on the validation set of the training OOD is no lower than that of OE. The motivation is that we want to be as good or better than OE for clean OOD detection while additionally having provable guarantees regarding adversarial robustness of OOD detection.
>
> *“Is there results for AAUC under different bias shift?”*
>
> Appendix D shows the trade-off curves of AUC, GAUC and accuracy on the validation sets for different shifts(Figure 3). AAUC is not shown in Figure 3 but the GAUC of ProoD is very close to its AAUC. Lower values of $\Delta$ lead to stronger guarantees, but usually weaken the clean AUCs.
>
> *“Highlight the best performing algorithm in each setup for table 2 and tables in the appendix”*
>
> We thank the reviewer for the suggestion and agree that highlighting can make our paper’s message more clear. Since our central aim is to provide models with high accuracy that also have provably adversarially robust OOD detection, we gray out all models that have an accuracy drop of more than 3% relative to the best performing model, since these would not be usable in practice. Of the remaining models we then highlight the best OOD detection performance. We have updated our Table 2 according to this scheme and propose to highlight all tables in this manner for the final version.
>
> *“Since binary shift is a core hyper-parameter for joint training, can authors explain why AUC increases with bias shift whereas GAUC decreases with the bias shift?”*
>
> Note that as we increase the bias shift, we reduce the importance of the binary discriminator. As $b\rightarrow \infty$, we recover OE, which has high clean AUC but zero GAUCs.
>
> *“Since the adversarial robustness is evaluated on OOD detection task instead of the naive classification, I am wondering if the authors explore the model-specific adaptive attack? [...] Particularly, I am not sure how robust the joint training procedure is against adaptive attacks.”*
>
> Note that our ensemble of attacks was already designed to be highly effective against our model. Concretely, the small gap between GAUCs and AAUCs shows that even a hypothetical perfect attack could not degrade any of ProoD’s AAUCs in Table 2 by more than 1.4% (the same cannot be said for any of the other methods). For classification it is standard to evaluate using AutoAttack, which also ensembles different methods and is known to be reliable. Out of those attacks APGD (an adaptive PGD) and SquareAttack (a black-box attack) can be adapted to our setting (maximizing confidence instead of a classification loss), and these attacks are in fact part of our ensemble of attacks which includes our own adaptive attacks. Compared to using only these standard attacks without our own adaptive attacks, we see that our ensemble is strictly stronger and on some datasets the gap is significant. For example, when attacking our ProoD model on CIFAR10 with OOD LSUN_CR, our attacks reach an AAUC of 59.7% (close to the lower bound 58.3%) whereas APGD+Square only reaches 73.4%.
>
> |              | CIFAR100 | SVHN | LSUN_CR | Smooth |
> |--------------|----------|------|---------|--------|
> |              | AAUC     | AAUC | AAUC    | AAUC   |
> | Robust       | 46.6     | 58.9 | 73.4    | 48.2   |
> | Robust-ProoD | 45.5     | 56.3 | 59.7    | 39.5   |
>
>
> Also note, that the joint-training procedure does not require any attack at all. Since we are not doing adversarial training but use IBP to compute bounds on our loss, attacks are only used at eval time.

---

> > ### Author Response · Authors · 2022-07-30
> > **Response to Reviewer vBCS (cont.)**
> >
> > *“I am wondering empirically what is the computation cost of the different variants of Prood, like Prood-S, Prood-disc, relative to other OOD detection methods, like OE?”*
> >
> > As we mention in lines 234-236, the computational cost of ProoD is only slightly higher than that of OE (less than 30% computational overhead). However, compared to other methods with adversarially robust OOD detection (like ACET and ATOM) our method is about an order of magnitude faster to train.
> >
> > *“As the authors mentioned in the paper, their approach only handles adversary around OOD samples instead of the in-distribution samples, and the studied adversary perturbation is generally limited to l_\infty ball. “, “The type of allowed adversary, limited to small perturbation sets, could be unrealistic for OOD data, which can have more different forms of uncertainty for the adversary.”*
> >
> > Please see the general response regarding robustness on the in-distribution. Regarding the perturbation model we agree that more general models are desirable, but we would like to emphasize that even these very small modifications are sufficient to yield high confidence predictions on out-distribution images. Thus we see our work as a first way to get certified robustness guarantees for out-of-distribution detection without harming standard performance.

---

### Official Review · Reviewer_7nMZ · 2022-07-12

**Rating:** 6
**Confidence:** 3
**Soundness:** 3 good
**Presentation:** 3 good
**Contribution:** 3 good

**Summary:**

This paper proposes a new OOD detection method that is provably robust against perturbations to OOD inputs. This method, called ProoD, has several desired properties including high detection performance as well as OOD detection certified robustness. ProoD makes use of a classifier and a certified binary discriminator for detection OOD samples and computes the class likelihoods as a product of the two functions. Theorem 1 in the paper guarantees that the network confidence would asymptotically converge to the uniform distribution and avoids overconfident predictions common in ReLU networks.

**Questions:**

I do not have any additional questions.

**Limitations:**

As it is also mentioned in the text, the proposed method only tackles one side of the problem which concerns adversarial robustness to OOD samples and by design does not consider the possibility of malicious perturbations to ID samples -- which is equally likely in principle. From that point of view the proposed method is severely handicapped.

**Strengths And Weaknesses:**

**strengths**
- well written paper
- detailed explanation of the methods, theoretical background and contribution
- thorough evaluations

**weaknesses**
- The main weakness of this work is the limited scope of the problem considered. As it is mentioned in the text, ProoD is designed to only consider the possibility of adversarial perturbations on the OOD samples (and not ID) and thus only seeks robustness in that regard. While this could be considered as a half-full half-empty kind of situation, I'm afraid in practice this is a critical downside which renders the method ultimately impractical.
- While the empirical results, reported in Table-2, are very thorough, the reported numbers are somewhat mixed. E.g. ProoD outperforms other baselines on most OOD sets in terms of the newly introduced GAUC. However, when considering the more common AAUC metric, it underperforms in most cases. Also related to the previous point, the evaluations seem to be always on OOD samples, so it limits the ability to assess if the method can additionally generalize its robustness  to ID samples.

---

> ### Author Response · Authors · 2022-07-30
> **Response to Reviewer 7nMZ**
>
> **We thank the reviewer for the detailed comments.**
>
> *“ProoD is designed to only consider the possibility of adversarial perturbations on the OOD samples (and not ID) and thus only seeks robustness in that regard. [...] I'm afraid in practice this is a critical downside which renders the method ultimately impractical.”*
>
> Please see our general response where we show that the combination of ProoD with an adversarially robust classifier on the in-distribution is adversarially robust on in- and out-distribution and has certified robustness guarantees on the out-distribution and even improves clean out-of-distribution detection. We think that this shows that ProoD is of high practical value. The reason why we did not discuss this directly in the paper is our emphasis on having certified guarantees without loss in prediction performance which is in general unavoidable when also requiring adversarial robustness on the in-distribution.
>
> *“However, when considering the more common AAUC metric, it underperforms in most cases.”*
>
> Please see our general response. We would like to add that papers on certified adversarial robustness on the in-distribution typically do not even compare to non-certified methods as the empirical adversarial robustness is in general much worse. We would also like to note that the original ATOM models reported very high AAUC which we could reduce to close to zero with our stronger attacks. In our point of view this shows nicely the high utility of certified robustness guarantees compared to a purely empirical evaluation.

---

### Author Response · Authors · 2022-07-30
**General remarks**

We thank all reviewers for their comments and constructive feedback. We also appreciate that the reviewers think that our paper is “well-written” (Reviewer 7nMZ, Reviewer ejdX), “well-motivated” (Reviewer vBCS), that our empirical evaluation is thorough (Reviewer 7nMZ, Reviewer ejdX, Reviewer vBCS) and that all appreciate our theoretical guarantees.

We want to address some general comments about the fact that our paper does not aim at adversarial robustness on the in-distribution as well as the fact that we do not necessarily outperform all other methods in all performance metrics.

The general motivation of our work is to show that certified guarantees regarding adversarial robustness on the out-distribution can be achieved without sacrificing standard accuracy and out-distribution detection performance. Currently, certified robust methods have not found their way into applications as the loss in clean performance cannot be tolerated.

In the following we discuss the two main points:

i) Adversarial robustness on the in-distribution: it is well known that in general adversarial robustness on the in-distribution comes at the price of reduced clean accuracy [Tsipras18]. As our goal is to maintain standard performance, it makes no sense for us to enforce it. However, we fully agree that it is reasonable to ask how our method would perform if we plug-in an adversarially robust model on the in-distribution as the classifier and this is why we added this experiment in Appendix K. As robust model we use the robust ResNet-18 from [Gowal21] (model from RobustBench [Croce21]) which has robust accuracy of 58.50% at $l_\infty$ of $\epsilon=8/255$ on the CIFAR10 test set (in-distribution) - denoted as Robust in the following. Robust-Prood is the combination of Robust with our Prood-Disc. As Equation (3) preserves the prediction, Robust-Prood has the same robust accuracy (and accuracy) as Robust on the in-distribution.
Moreover, the table below shows that Robust-Prood has the same or better clean AUCs as well as AAUCs than Robust and additionally it now also has non-trivial adversarial robustness guarantees (GAUCs) on the out-distribution. Thus, ProoD also helps in the case where adversarial robustness on the in-distribution is required, which shows how versatile our approach is.

|              |      |      | CIFAR100 |      |      | SVHN |      |      | LSUN |      |      | Smooth |      |
|--------------|------|------|----------|------|------|------|------|------|------|------|------|--------|------|
|              | Acc  | AUC  | GAUC     | AAUC | AUC  | GAUC | AAUC | AUC  | GAUC | AAUC | AUC  | GAUC   | AAUC |
| Robust       | 87.4 | 82.7 | 0.0      | 75.3 | 90.0 | 0.0  | 84.0 | 89.7 | 0.0  | 82.9 | 94.5 | 0.0    | 85.7 |
| Robust-ProoD | 87.4 | 82.9 | 14.3     | 75.6 | 90.4 | 17.9 | 84.5 | 90.1 | 18.8 | 83.4 | 94.4 | 10.3   | 85.8 |

ii) Performance Comparison: Reviewers 7nMZ and ejdX note that ProoD does not always outperform the other methods in all performance metrics e.g. in AAUC. However, first we would like to note that the AAUC of Robust-Prood would outperform the AAUCs of any other method on CIFAR10. Second, and more importantly, our goal is to get guarantees for robust OOD detection *while* maintaining standard accuracy and clean OOD detection performance. It is true that in some cases GOOD outperforms  ProoD in terms of GAUC and AAUC but it has more than 7.6% worse clean accuracy on CIFAR10 which we consider not acceptable for a practical application. Similarly, for CIFAR100 ATOM outperforms ProoD in terms of AAUC but has 8.8% worse clean accuracy which is not acceptable. So among the methods with similar clean accuracy as OE/plain, ProoD performs similarly or better in terms of AUC, AAUC and is  the only method having GAUCs. Apart from this we note that GOOD, ATOM and ACET all suffer from training instabilities.

We think that ProoD shows how certified techniques can have an impact in practical applications.

[Tsipras18] D. Tsipras, S. Santurkar, L. Engstrom, A. Turner, A. Madry “Robustness may be at odds with accuracy” at ICLR’18

[Gowal21] S. Gowal, S. Rebuffi, O. Wiles, F.Stimberg, D. Calian, T. Mann “Improving Robustness using Generated Data”, NeurIPS’21

[Croce21] F. Croce, M. Andriushchenko, V. Sehwag, E. Debenedetti, N. Flammarion, M. Chiang, P. Mittal, M. Hein “RobustBench: a standardized adversarial robustness benchmark”, NeurIPS’21 Dataset and Benchmark track

---

### Meta-Review · Area_Chair_gfnj · 2022-08-26

**Recommendation:** Accept
**Confidence:** Certain

**Metareview:**

The paper considers the task of OOD detection facing adversarial manipulations. It shows two existing defenses can be broken. It then proposes a method called ProoD to construct a classifier with both provability and high clean accuracy, and prove that it avoids asymptotic overconfidence.

The paper studies an important topic and has made significant contributions: well-designed method with guarantees, thorough experiments with strong performance. The reviewers have some concerns that are addressed by the reviewers:

1. Interpretation of the experimental results. ProoD doesn't always outperformance in all metrics. In the response, the authors clarify that among the methods with high accuracy, ProoD enjoys strong out-distribution robustness and also with certificates. Making this explicit can improve the presentation.

2. Presentation of the experimental results. The authors have incorporated some suggestions and the presentation is improved.

Overall, the work is a good contribution, and acceptance is recommended.


**Award:**

No

---

### Decision · Program_Chairs · 2022-09-14

Accept